# Antibiotics promote intestinal growth of carbapenem-resistant *Enterobacteriaceae* by enriching nutrients and depleting microbial metabolites

Alexander Y. G. Yip [1], Olivia G. King[2], Oleksii Omelchenko [1], Sanjana Kurkimat[1], Victoria Horrocks [1], Phoebe Mostyn[1], Nathan Danckert[3], Rohma Ghani[3,4], Giovanni Satta[5], Elita Jauneikaite [6,7], Frances J. Davies[4], Thomas B. Clarke [2], Benjamin H. Mullish [3,8], Julian R. Marchesi[3] & Julie A. K. McDonald [1] ✉

The intestine is the primary colonisation site for carbapenem-resistant *Enterobacteriaceae* (CRE) and serves as a reservoir of CRE that cause invasive infections (e.g. bloodstream infections). Broad-spectrum antibiotics disrupt colonisation resistance mediated by the gut microbiota, promoting the expansion of CRE within the intestine. Here, we show that antibiotic-induced reduction of gut microbial populations leads to an enrichment of nutrients and depletion of inhibitory metabolites, which enhances CRE growth. Antibiotics decrease the abundance of gut commensals (including *Bifidobacteriaceae* and *Bacteroidales*) in ex vivo cultures of human faecal microbiota; this is accompanied by depletion of microbial metabolites and enrichment of nutrients. We measure the nutrient utilisation abilities, nutrient preferences, and metabolite inhibition susceptibilities of several CRE strains. We find that CRE can use the nutrients (enriched after antibiotic treatment) as carbon and nitrogen sources for growth. These nutrients also increase in faeces from antibiotic-treated mice and decrease following intestinal colonisation with carbapenem-resistant *Escherichia coli*. Furthermore, certain microbial metabolites (depleted upon antibiotic treatment) inhibit CRE growth. Our results show that killing gut commensals with antibiotics facilitates CRE colonisation by enriching nutrients and depleting inhibitory microbial metabolites.

Antibiotic resistance presents a serious threat to human health, resulting in treatment failures, infection relapses, longer hospitalisations, poor clinical outcomes, and increased healthcare costs[1]. Treatment options are increasingly limited, less effective, and may involve administration of antibiotics that are more toxic to the patient[2]. There is an urgent need to develop new treatments for infections by antibiotic-resistant bacteria. In particular, alternative approaches are required to prevent the development of invasive infections by

extremely drug-resistant pathogens such as carbapenem-resistant *Enterobacteriaceae* (CRE), which present an urgent public health threat[2]. However, there are many difficulties associated with the development of new antibiotics to treat these multidrug-resistant infections[2].

The intestine is the primary colonisation site for CRE and serves as a reservoir of CRE that seed difficult-to-treat invasive infections (such as bloodstream infections and recurrent urinary tract infections)[3].

Therefore, one approach to prevent the development of invasive CRE infections would be to prevent CRE from colonising the intestine, or to decolonise patients with pre-existing CRE intestinal colonisation. However, to do so we must first understand what drives CRE colonisation in the intestines of susceptible patients.

Healthy gut microbiota exhibit colonisation resistance, where gut commensals prevent pathogens from colonising the intestine. However, it is well established that broad spectrum antibiotics disrupt colonisation resistance and significantly promote the expansion of pathogens within the intestine[4]. In particular, carbapenems, piperacillin/tazobactam, ciprofloxacin, and cephalosporins are known to promote CRE intestinal colonisation[5]. The mechanisms of colonisation resistance that healthy gut microbiota use to protect against CRE intestinal colonisation are not fully understood. However, based on findings from other pathogens, nutrient competition and metabolite inhibition are expected to play an important role[6–10].

Bacteria must have access to nutrients that support their growth to successfully colonise the intestine. Different bacteria have different nutrient utilisation abilities, and the diversity and concentration of the available nutrients will impact their growth[11]. Although bacteria may be able to utilise a variety of nutrients, they generally have a prioritised utilisation of nutrients that varies between different bacterial species[12]. However, nutrients are limited within the intestine as bacteria with similar or overlapping nutrient utilisation abilities will compete for these nutrients and occupy similar niches[13]. The nutrient-niche hypothesis predicts that antibiotic-mediated disruption of the gut microbiota leads to reduced competition for nutrients and an increase in nutrient availability in the gut, which could promote *Enterobacteriaceae* growth[14].

Nutrient metabolism by the gut microbiota results in the production of metabolites, some of which can inhibit pathogen growth[7,8]. Polysaccharide fermentation results in the production of metabolites such as short chain fatty acids (SCFAs; e.g. formate, acetate, propionate, butyrate, valerate) and organic acids (e.g. succinate, lactate, ethanol)[15]. Protein fermentation results in the production of SCFAs and branched chain fatty acids (BCFAs; e.g. isobutyrate and isovalerate)[15]. However, killing gut commensals with antibiotics can decrease the production of microbial metabolites and enable pathogen growth[7,16].

Understanding how broad-spectrum antibiotics disrupt colonisation resistance to promote CRE growth will facilitate the rational design of microbiome therapeutics to prevent or treat CRE intestinal colonisation, and therefore prevent the subsequent development of difficult-to-treat invasive CRE infections. However, we lack data demonstrating how clinically relevant antibiotics (that promote CRE intestinal colonisation in humans) impact nutrient availability and metabolite production in the gut microbiome. We also lack data demonstrating which nutrients are utilised by CRE in an antibiotic-treated gut microbiome. This data is essential to develop effective microbiome therapeutics, as microbiome therapeutics that only deplete a subset of the available nutrients that CRE can utilise will have limited clinical impact to prevent or restrict CRE growth. A small number of microbial metabolites have been proposed to inhibit CRE intestinal colonisation[16,17]. However, given the severe impact of broad-spectrum antibiotics on the gut microbiota, we expected that these antibiotics would have a much more significant impact on microbial metabolism in the intestine. Therefore, we hypothesised that broad-spectrum antibiotics that promote CRE growth significantly enrich for a wide range and diversity of nutrients and deplete a wide range and diversity of metabolites, resulting in a significant change in the nutrient and metabolite landscape CRE encounter in antibiotic-treated gut microbiota and creating a niche that promotes CRE growth.

In this study we demonstrated that broad-spectrum antibiotics enriched for a wide range of nutrients in the intestine, generating a nutrient-enriched niche that supported CRE growth. We showed that these nutrients could act as carbon and nitrogen sources to support CRE growth, where CRE strains showed an ordered preference for specific nutrients. Oxygen availability is also increased in an antibiotic-treated gut, and we showed that CRE growth was higher on these nutrients in the presence of oxygen. We also demonstrated that these broad-spectrum antibiotics depleted a wide range of metabolites and generated a metabolite-depleted niche that no longer inhibited CRE growth. We showed that microbial metabolites (that were decreased with antibiotics) were inhibitory towards CRE growth. Together, these results demonstrated that killing gut commensals with antibiotics reduces colonisation resistance by disrupting nutrient metabolism. This results in an intestinal niche that supports the expansion of CRE by enriching nutrients that support CRE growth and depleting metabolites that inhibit CRE growth. Our findings will significantly impact the design of microbiome therapeutics to treat CRE intestinal colonisation. Bacterial consortia that compose microbiome therapeutics must fully occupy the same nutrient-defined intestinal niches that CRE occupy in the gut to effectively prevent or restrict CRE intestinal growth.

## Results

### Antibiotics decreased the abundance of important bacterial families in faecal microbiota

We performed ex vivo faecal culture experiments to measure the effects of eight broad-spectrum antibiotics on the faecal microbiota from 11 healthy human donors. Antibiotics tested included meropenem (MEM), imipenem/cilastatin (IPM), ertapenem (ETP), piperacillin/tazobactam (TZP), ciprofloxacin (CIP), ceftriaxone (CRO), ceftazidime (CAZ), and cefotaxime (CTX). These antibiotics are frequently used clinically and known to promote the intestinal colonisation with CRE[5]. Faecal cultures are highly reproducible, allowing for the parallel testing of multiple groups that are inoculated from a single faecal sample[18]. For each experiment a faecal sample was inoculated into a distal gut growth medium supplemented with one of the eight antibiotics or not supplemented with antibiotics (antibiotic-naïve control). First, we measured how these different antibiotics (that promote susceptibility to CRE intestinal colonisation) impacted the abundance of gut commensal taxa from faecal microbiota.

We found that antibiotics decreased the absolute abundance of several bacterial families, in particular *Bifidobacteriaceae* and families from the order *Bacteroidales* (Fig. 1a). Carbapenems decreased the abundance of *Bifidobacteriaceae*, *Coriobacteriaceae*, *Bacteroidaceae*, *Barnesiellaceae*, *Butyricicoccaceae*, and other bacterial families. TZP decreased the abundance of *Bifidobacteriaceae* and *Bacteroidaceae*. CIP decreased the abundance of *Bifidobacteriaceae*, *Barnesiellaceae* and *Sutterellaceae*. Cephalosporins decreased the abundance of *Bifidobacteriaceae* only.

Overall, antibiotic treatment resulted in a decrease in the abundance of several bacterial families in faecal microbiota, in particular *Bifidobacteriaceae* and families from the *Bacteroidales* order.

### Antibiotics enriched nutrients and depleted microbial metabolites in faecal microbiota

Antibiotic-mediated killing of gut commensals can reduce competition for nutrients, resulting in an increased availability of nutrients that could support pathogen growth. Antibiotics can also decrease the concentration of microbial metabolites which may be inhibitory to pathogen growth. However, we do not know which nutrients and metabolites are impacted by treatment with different antibiotics that promote the intestinal colonisation with CRE. Therefore, we also measured the effects of the different antibiotics on nutrient availability and metabolite production in antibiotic-treated and antibiotic-naïve faecal cultures using $^1$H-NMR spectroscopy.

Antibiotic treatment of faecal microbiota increased the concentration of several nutrients, including monosaccharides (arabinose, fructose, fucose, galactose, glucose, mannose, ribose,

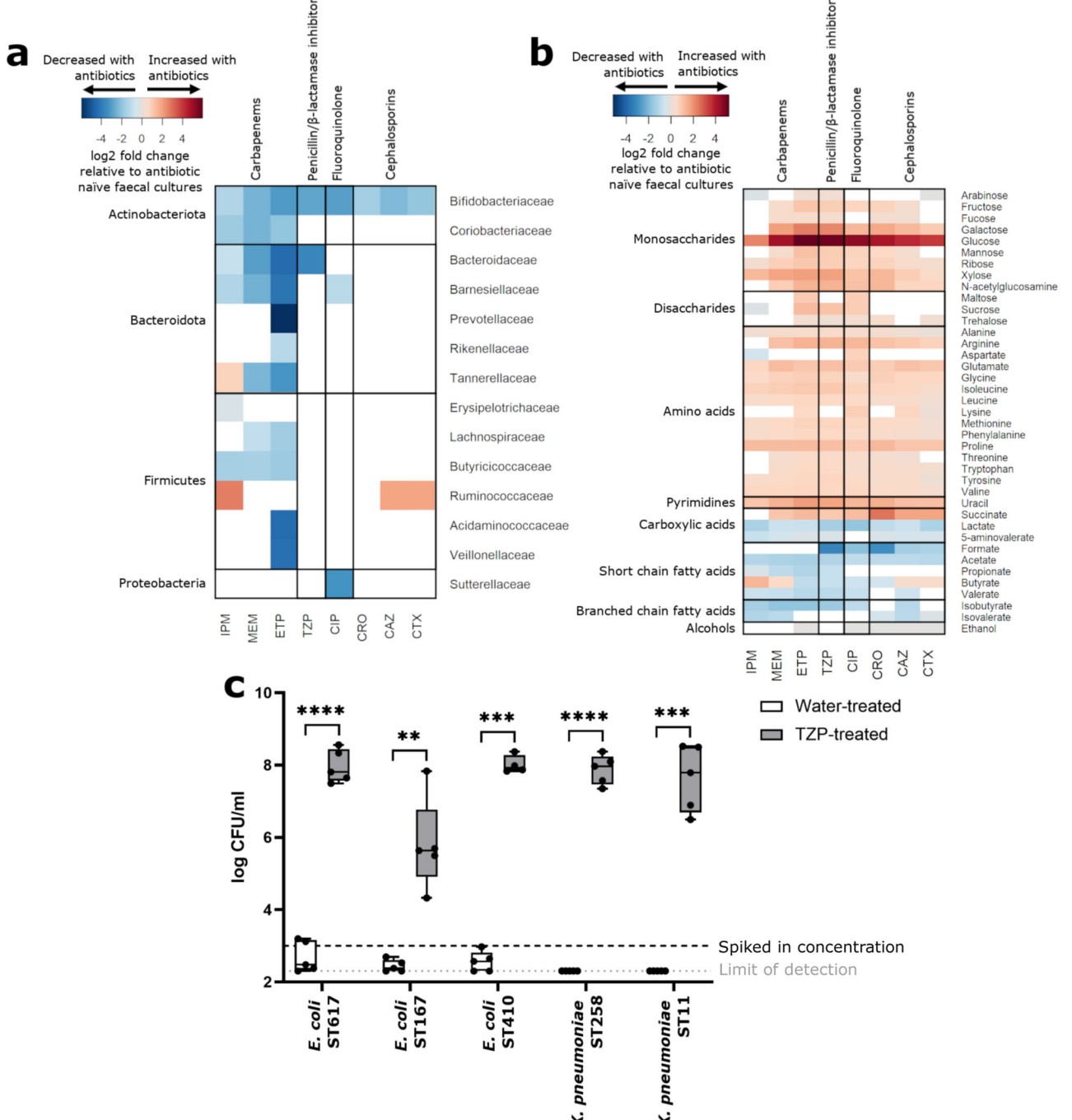

**Fig. 1 | CRE growth is promoted in antibiotic-treated human faecal microbiota that had a significant decrease in the abundance of several bacterial families, a significant increase in the concentration of nutrients, and a significant decrease in the concentration of microbial metabolites. a** Log2 fold change in bacterial families significantly decreased with antibiotics (blue) or significantly increased with antibiotics (red), relative to antibiotic-naïve faecal cultures. Wilcoxon signed rank test (two-sided) of log-transformed abundances with Benjamini Hochberg false discovery rate (FDR) correction, $p < 0.05$. Bacterial families not significantly changed with antibiotics were not plotted (white). $n = 11$ healthy human faecal donors. **b** Log2 fold change in nutrients and metabolites significantly increased with antibiotics (red) or significantly decreased with antibiotics (blue), relative to antibiotic-naïve faecal cultures. Wilcoxon signed rank test (two-sided) with Benjamini Hochberg FDR, $p < 0.05$. Nutrients and metabolites not significantly

changed with antibiotics were not plotted (white). $n = 11$ healthy human faecal donors. **c** TZP significantly promoted CRE growth in human faecal microbiota. Each CRE strain was spiked into TZP-treated or water-treated faecal cultures at $10^3$ CFU/ml at the start of the experiment (represented by the black horizontal dashed line) and CRE growth was quantified using selective plate counts. Limit of detection represented by the grey horizontal dotted line. The box indicates the median as a line in the box and excludes the upper and lower 25% (quartiles) of data, and the whiskers extend to the maximum and minimum values. $n = 5$ healthy human faecal donors, paired t-test (two-sided) of log transformed CFU/ml, ** = $P \leq 0.01$, *** = $P \leq 0.001$, **** = $P \leq 0.0001$. IPM Imipenem/cilastatin, MEM meropenem, ETP ertapenem, TZP piperacillin/tazobactam, CIP ciprofloxacin, CRO ceftriaxone, CAZ ceftazidime, CTX cefotaxime. Source data and $P$ values are provided in a Source Data file.

xylose, N-acetylglucosamine), disaccharides (maltose, sucrose, trehalose), amino acids (alanine, arginine, aspartate, glutamate, glycine, isoleucine, leucine, lysine, methionine, phenylalanine, proline, threonine, tryptophan, tyrosine, valine), uracil, and succinate (Fig. 1b). Antibiotics also decreased the concentration of microbial metabolites, including SCFAs (formate, acetate, propionate, valerate), BCFAs (isobutyrate, isovalerate), lactate, 5-aminovalerate, and ethanol (Fig. 1b). Butyrate decreased in ETP, TZP, CIP and CRO-treated faecal microbiota but increased in MEM, IPM, CAZ, and CTX-treated faecal microbiota.

We spiked CRE isolates into antibiotic-treated and antibiotic-naïve faecal cultures to determine whether the increase in nutrient availability and decrease in the concentration of microbial metabolites led to environmental conditions that promoted CRE growth. We tested TZP in this experiment because TZP has been demonstrated to promote CRE intestinal colonisation in humans and mice[5,19,20]. We also showed that changes in bacterial families, nutrient availability, and microbial metabolites measured in TZP-treated faecal microbiota were representative of changes measured in faecal microbiota treated with the other antibiotics tested in this study (Fig. 1a, b). We demonstrated that carbapenem-resistant *E. coli* and *K. pneumoniae* growth were significantly promoted in TZP-treated faecal cultures and suppressed in antibiotic-naïve faecal cultures (Fig. 1c).

Although specific changes were measured for each antibiotic that was tested, overall antibiotic treatment of a faecal microbiota resulted in an increase in monosaccharides (in particular glucose), disaccharides, amino acids, uracil, and succinate and a decrease in the concentration of SCFAs, BCFAs, lactate, 5-aminovalerate, and ethanol. This created a nutrient-enriched niche that supported CRE growth and a metabolite-depleted niche that no longer inhibited CRE growth.

## Bacterial taxa that were decreased with antibiotics were positively correlated with microbial metabolites and negatively correlated with nutrients

Regularised canonical correlation analysis (rCCA) is an unsupervised exploratory approach used to analyse the correlation structure between two multivariate datasets that have been acquired from the same samples[21,22]. We used rCCA modelling to measure correlations between 16S rRNA gene sequencing data and ¹H-NMR spectroscopy data from antibiotic-treated and antibiotic-naïve faecal cultures. The aim of this analysis was to link changes in bacterial families to changes in nutrients and metabolites following antibiotic treatment, to provide targets for the future development of microbiome therapeutics to restore colonisation resistance against CRE.

Unit representation plots show sample clustering projected into the *XY*-variate space (in this study, *X* and *Y* represent the 16S rRNA sequencing dataset and ¹H-NMR spectroscopy dataset)[21]. The unit representation plots showed a separation between antibiotic-treated and antibiotic-naïve faecal cultures along the first canonical variate, indicating antibiotic treatment caused a separation in these groups based on the correlation structure of bacterial taxa and nutrient/metabolite profiles (Supplementary Fig. S1).

Correlation circle plots show the correlation structure between variables and the canonical variates, highlighting the important variables amongst the noisy variables[21,22]. In these plots, variables projected in the same direction from the origin have a positive correlation, and variables projected in opposite directions from the origin have a negative correlation. Variables sitting at farther distances from the origin have stronger correlations than variables sitting closer to the origin (variables with negligible correlations were not plotted). In this study, correlation circle plots showed moderate and strong negative correlations between bacterial families that were decreased with antibiotics and monosaccharides, disaccharides, amino acids, uracil, and succinate (Fig. 2). They also showed moderate and strong positive correlations between bacterial families that were decreased with antibiotics and SCFAs, BCFAs, 5-aminovalerate, and ethanol (Fig. 2).

Partial Spearman correlations were also calculated to determine significant correlations between the ¹H-NMR spectroscopy data and 16S rRNA gene sequencing data and showed similar results to the data presented in the correlation circle plots (Supplementary Figs. S2–S9).

Therefore, antibiotic treatment (using antibiotics that promote CRE intestinal colonisation) resulted in a reduction of bacterial families that are positively correlated with metabolites (indicating that these bacterial families may produce these metabolites) and negatively correlated with nutrients (indicating that these bacterial families may consume these nutrients). This highlights the potential role of these gut commensals for providing colonisation resistance against CRE in antibiotic-naïve faecal microbiota.

## CRE can grow using nutrients that were elevated in antibiotic-treated faecal microbiota

We demonstrated that monosaccharides, disaccharides, amino acids, uracil, and succinate were elevated with antibiotic treatment of faecal microbiota (Fig. 1b). Our next aim was to demonstrate that these nutrients could support CRE growth. The growth of carbapenem-resistant *Escherichia coli*, *Klebsiella pneumoniae*, and *Enterobacter hormaechei* strains were measured in a mixture of these nutrients supplemented into M9 minimal medium (lacking any other carbon or nitrogen sources) and changes in $OD_{600}$ were measured over time. Antibiotic-mediated disruption of the gut microbiota significantly increases oxygen availability in the gut, which may allow *Enterobacteriaceae* to grow better on available carbon sources[14]. Therefore, we investigated CRE growth in both the presence and absence of oxygen.

We showed that CRE growth was supported by the nutrient mixture, where these nutrients were acting as the sole carbon and nitrogen sources (Fig. 3). For each CRE isolate growth was significantly higher under aerobic conditions compared to anaerobic conditions.

Therefore, CRE can grow on nutrients that were elevated with antibiotics, and increased oxygen availability promotes higher levels of CRE growth on these nutrients. This led us to conclude that the significant increase of these nutrients following antibiotics contributes to the loss of colonisation resistance against CRE.

## CRE can use monosaccharides, disaccharides and amino acids as carbon sources to support their growth

Above we demonstrated that a mixture of monosaccharides, disaccharides, amino acids, uracil, and succinate were able to support CRE growth in a minimal medium (Fig. 3). However, it was not clear which specific nutrients were utilised by each CRE isolate. Therefore, we next tested which individual nutrients could act as sole carbon sources to support CRE growth. The carbon utilisation abilities of carbapenem-resistant *E. coli*, *K. pneumoniae* and *E. hormaechei* strains were demonstrated by adding a single carbon source into M9 minimal medium and monitoring changes in $OD_{600}$ over time.

When grown anaerobically, *E. coli*, *K. pneumoniae* and *E. hormaechei* could use most monosaccharides and disaccharides as carbon sources to support their growth (Fig. 4), with some carbon sources supporting higher levels of growth than others (Supplementary Fig. S10). *E. coli* could grow on arabinose, fructose, galactose, glucose, mannose, xylose, N-acetylglucosamine, and trehalose. *K. pneumoniae* could grow on arabinose, fructose, galactose, glucose, mannose, xylose, N-acetylglucosamine, maltose, sucrose and trehalose. *E. hormaechei* could grow on arabinose, fructose, galactose, glucose, mannose, ribose, xylose, N-acetylglucosamine, maltose, sucrose, and trehalose. Carbon sources that did not support *E. coli*, *K. pneumoniae* and *E. hormaechei* growth under anaerobic conditions are shown in Supplementary Fig. S11.

When grown aerobically, *E. coli*, *K. pneumoniae*, and *E. hormaechei* could use the same carbon sources that were used anaerobically plus additional carbon sources to support their growth (Fig. 4). Again, some

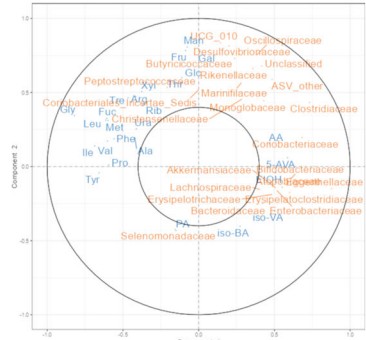

IPM-treated vs antibiotic-naive

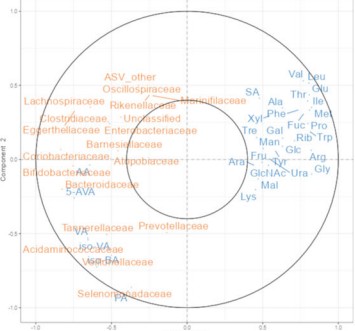

MEM-treated vs antibiotic-naive

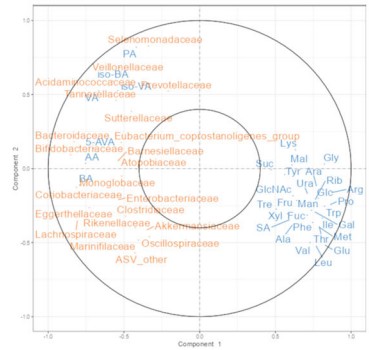

ETP-treated vs antibiotic-naive

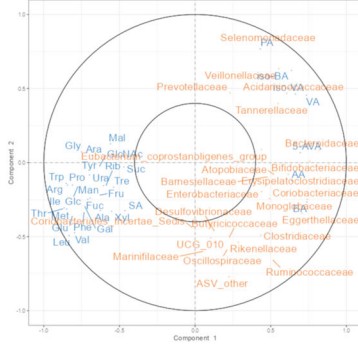

TZP-treated vs antibiotic-naive

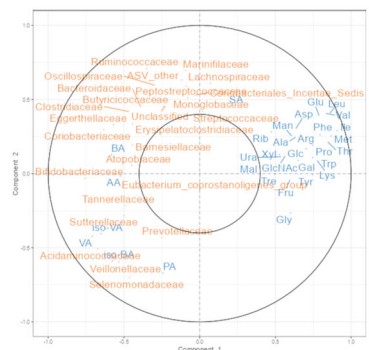

CIP-treated vs antibiotic-naive

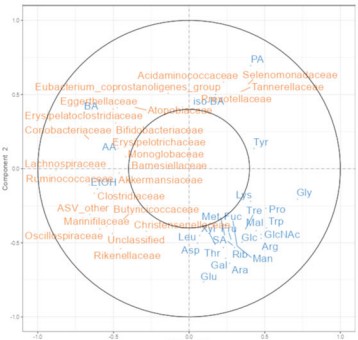

CRO-treated vs antibiotic-naive

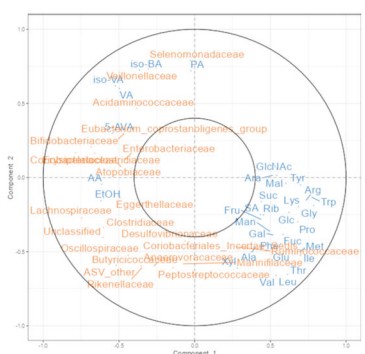

CAZ-treated vs antibiotic-naive

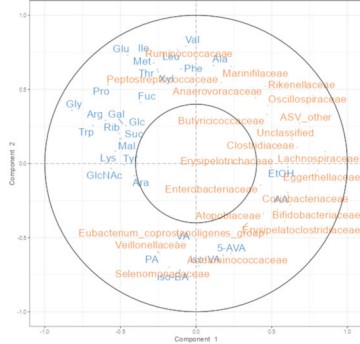

CTX-treated vs antibiotic-naive

**Fig. 2 | Bacterial families that were decreased in antibiotic-treated human faecal microbiota were positively correlated with microbial metabolites and negatively correlated with nutrients.** rCCA model correlating 16S rRNA gene sequencing data (family level) and ¹H-NMR spectroscopy data (n = 11 healthy faecal donors). Correlation circle plot showing correlations between variables from antibiotic-treated and antibiotic-naïve samples. Nutrients and metabolites are shown in blue and bacterial families are shown in orange. The following abbreviations are used for nutrients and metabolites: Ara arabinose, Fru fructose, Fuc fucose, Gal galactose, Glc glucose, Man mannose, Rib ribose, Xyl xylose, GlcNAc N-

acetylglucosamine, Mal maltose, Suc sucrose, Tre trehalose, Ala alanine, Arg arginine, Asp aspartate, Glu glutamate, Gly glycine, Ile isoleucine, Leu leucine, Lys lysine, Met methionine, Phe phenylalanine, Pro proline, Thr threonine, Trp tryptophan, Tyr tyrosine, Val valine, Ura uracil, SA succinate, LA lactate, 5-AVA 5-aminovalerate, FA formate, AA acetate, PA propionate, BA butyrate, VA valerate, iso-BA isobutyrate, iso-VA isovalerate, EtOH ethanol, IPM Imipenem/cilastatin, MEM meropenem, ETP ertapenem, TZP piperacillin/tazobactam, CIP ciprofloxacin, CRO ceftriaxone, CAZ ceftazidime, CTX cefotaxime. Source data are provided as a Source Data file.

carbon sources supported higher levels of growth than others (Supplementary Fig. S10). *E. coli* could grow on arabinose, fructose, galactose, glucose, mannose, ribose, xylose, N-acetylglucosamine, maltose, sucrose, trehalose and alanine. *K. pneumoniae* could grow on arabinose, fructose, galactose, glucose, mannose, ribose, xylose, N-

acetylglucosamine, maltose, sucrose, trehalose, alanine, glutamate and proline. *E. hormaechei* could grow on arabinose, fructose, fucose, galactose, glucose, mannose, ribose, xylose, N-acetylglucosamine, maltose, sucrose, trehalose, alanine and glutamate. Carbon sources that did not support *E. coli*, *K. pneumoniae*, and *E. hormaechei* growth under aerobic conditions are shown in Supplementary Fig. S11.

Next, we compared the growth of two NDM *E. coli* strains (ST617 and ST167) on these carbon sources to determine whether carbon utilisation abilities varied between different strains of the same species. The carbon utilisation abilities of *E. coli* ST617 and *E. coli* ST167 were overall very similar, but there were some differences between these two strains (Supplementary Figs. S12, S13). When grown anaerobically, *E. coli* ST167 could grow on arabinose, fructose, galactose, glucose, mannose, ribose, xylose, N-acetylglucosamine, maltose, sucrose, and trehalose. *E. coli* ST617 could use these same carbon sources as *E. coli* ST167, except that *E. coli* ST617 was unable to use ribose, maltose, and sucrose. When grown aerobically, *E. coli* ST167 could grow on arabinose, fructose, fucose, galactose, glucose, mannose, ribose, xylose, N-acetylglucosamine, maltose, sucrose, trehalose, alanine and succinate. Again, *E. coli* ST617 could use these same carbon sources as *E. coli* ST167, except that *E. coli* ST617 was unable to use fucose and succinate.

Overall, these carbapenem-resistant *E. coli*, *K. pneumoniae*, and *E. hormaechei* strains were able to utilise individual nutrients as carbon sources to support their growth. However, each isolate was able to grow to higher levels when grown aerobically versus anaerobically.

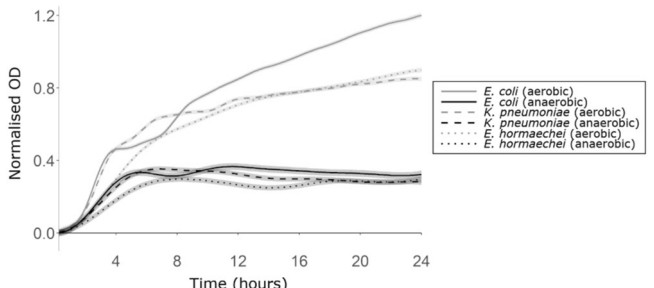

**Fig. 3 | CRE growth was supported by a mixture of nutrients that were elevated in antibiotic-treated faecal microbiota.** AMiGA-predicted growth curves for *E. coli* ST617, *K. pneumoniae* ST1026, and *E. hormaechei* ST278 grown on M9 minimal medium supplemented with a mixture of nutrients under anaerobic or aerobic conditions. Growth of each isolate was measured with six replicates in two independent experiments. The predicted mean of growth is shown with bold lines and the predicted 95% credible intervals are shown with the shaded bands. OD Optical density. Source data are provided as a Source Data file.

**Fig. 4 | CRE growth was supported by many individual carbon sources that were elevated in antibiotic-treated faecal microbiota.** AMiGA-predicted growth curves for *E. coli* ST617, *K. pneumoniae* ST1026, and *E. hormaechei* ST278 grown on M9 minimal medium supplemented with a single carbon source (or water as the no carbon control) under anaerobic or aerobic conditions. Growth of each isolate was measured with 6 replicates in 2–3 independent experiments. The predicted mean of growth is shown with bold lines and the predicted 95% credible intervals are shown with the shaded bands. OD Optical density. Source data are provided as a Source Data file.

Moreover, each isolate was also able to grow on a larger number of carbon sources aerobically versus anaerobically.

## CRE can use amino acids, N-acetylglucosamine and uracil as nitrogen sources to support their growth

Above we showed that most amino acids and uracil were not used as carbon sources by CRE (Supplementary Fig. S11). However, amino acids, N-acetylglucosamine, and uracil may instead act as nitrogen sources to support CRE growth. Therefore, we next tested the nitrogen utilisation abilities of carbapenem-resistant *E. coli*, *K. pneumoniae*, and *E. hormaechei* by adding a single nitrogen source into M9 minimal medium and monitoring changes in $OD_{600}$ over time.

When grown anaerobically, *E. coli*, *K. pneumoniae* and *E. hormaechei* could use some amino acids and N-acetylglucosamine as nitrogen sources to support their growth (Fig. 5), with some nitrogen sources supporting higher levels of growth than others (Supplementary Fig. S14). *E. coli* could grow on arginine, aspartate, valine and N-acetylglucosamine. *K. pneumoniae* and *E. hormaechei* could grow on arginine, aspartate and N-acetylglucosamine.

When grown aerobically, *E. coli*, *K. pneumoniae* and *E. hormaechei* could use the same nitrogen sources that were used anaerobically plus additional nitrogen sources to support their growth (Fig. 5). Again, some nitrogen sources supported higher levels of growth than others (Supplementary Fig. S14). *E. coli* could grow on alanine, arginine, aspartate, glutamate, glycine, isoleucine, leucine, lysine, methionine, proline, threonine, tryptophan, tyrosine, valine, N-acetylglucosamine and uracil. *K. pneumoniae* could grow on alanine, arginine, aspartate, glutamate, glycine, isoleucine, leucine, lysine, methionine, proline, and uracil. *E. hormaechei* could grow on alanine, arginine, aspartate, glutamate, glycine, isoleucine, leucine, lysine, methionine, phenylalanine, proline, threonine, N-acetylglucosamine and uracil.

Again, we compared the growth of two NDM *E. coli* strains (ST617 and ST167) on these nitrogen sources to determine whether nitrogen utilisation abilities varied between different strains of the same species. The nitrogen utilisation abilities of *E. coli* ST617 and *E. coli* ST167 were overall very similar, but there were some differences between these two strains (Supplementary Figs. S15, S16). When grown anaerobically, *E. coli* ST167 could grow on aspartate and N-acetylglucosamine. *E. coli* ST617 could use these same nitrogen sources as *E. coli* ST167, except that *E. coli* ST617 was also able to use arginine and valine (at low levels). When grown aerobically, *E. coli* ST167 could grow on alanine, arginine, aspartate, glutamate, glycine, isoleucine, leucine, lysine, methionine, phenylalanine, proline, threonine, tryptophan, valine, and N-acetylglucosamine. Again, *E. coli* ST617 could use these same nitrogen sources as *E. coli* ST167, except that *E. coli* ST617 was unable to use phenylalanine but was able to use tyrosine and uracil (at low levels).

Overall, these carbapenem-resistant *E. coli*, *K. pneumoniae* and *E. hormaechei* strains were able to utilise individual nutrients as nitrogen sources to support their growth. Each isolate was able to grow to higher levels when grown aerobically versus anaerobically. Moreover, each isolate was also able to grow on a larger number of nitrogen sources aerobically versus anaerobically.

## CRE had an ordered preference of nutrients when provided with a mixture of nutrients

Above we demonstrated that CRE can use monosaccharides, disaccharides, and some amino acids as carbon sources and amino acids, N-acetylglucosamine, and uracil as nitrogen sources to support their growth. However, carbon utilisation assays were carried out using a single nitrogen source and nitrogen utilisation assays were carried out using a single carbon source. As we demonstrated in Fig. 1b, there are

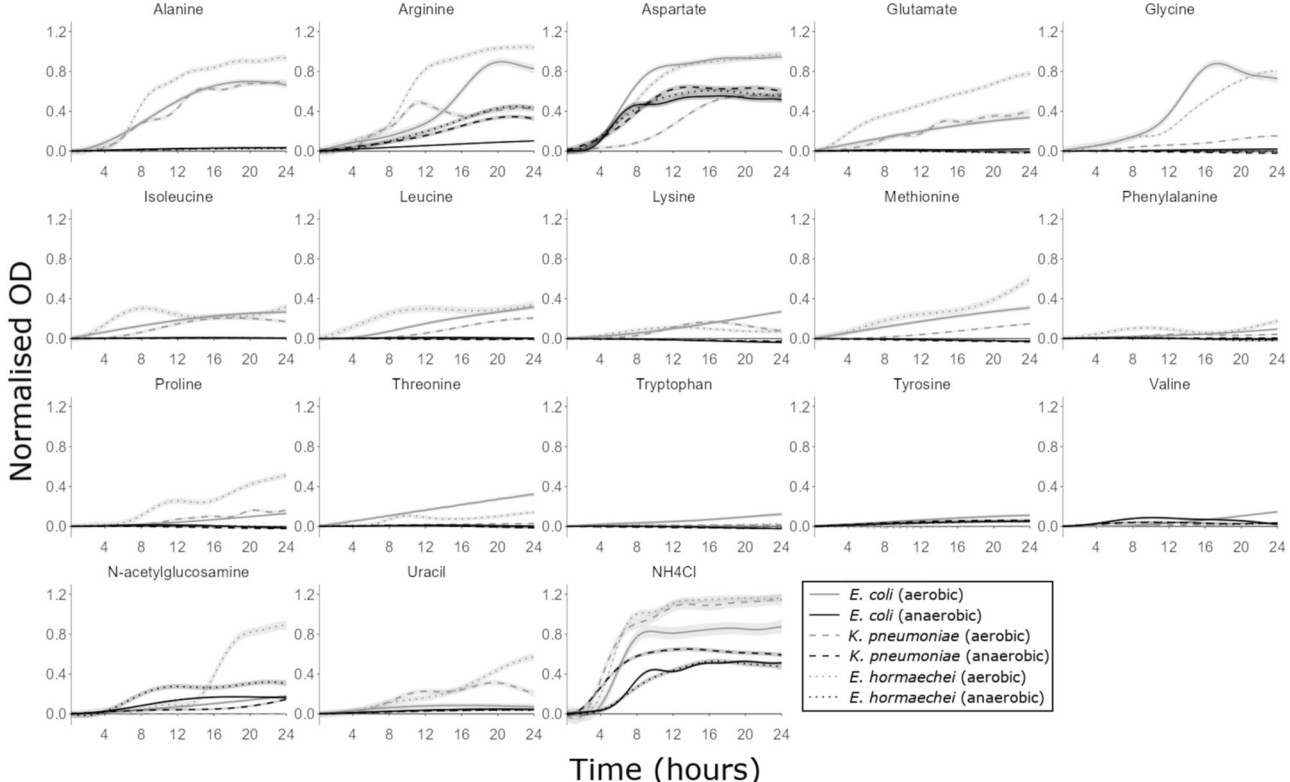

**Fig. 5 | CRE growth was supported by individual nitrogen sources that were elevated in antibiotic-treated faecal microbiota.** AMiGA-predicted growth curves for *E. coli* ST617, *K. pneumoniae* ST1026 and *E. hormaechei* ST278 grown on M9 minimal medium supplemented with a single nitrogen source (or water as the no nitrogen control) under anaerobic or aerobic conditions. Growth of each isolate was measured with six replicates in 2–3 independent experiments. The predicted mean of growth is shown with bold lines and the predicted 95% credible intervals are shown with the shaded bands. NH4Cl was used as a positive control for growth as a sole nitrogen source. OD Optical density. Source data are provided as a Source Data file.

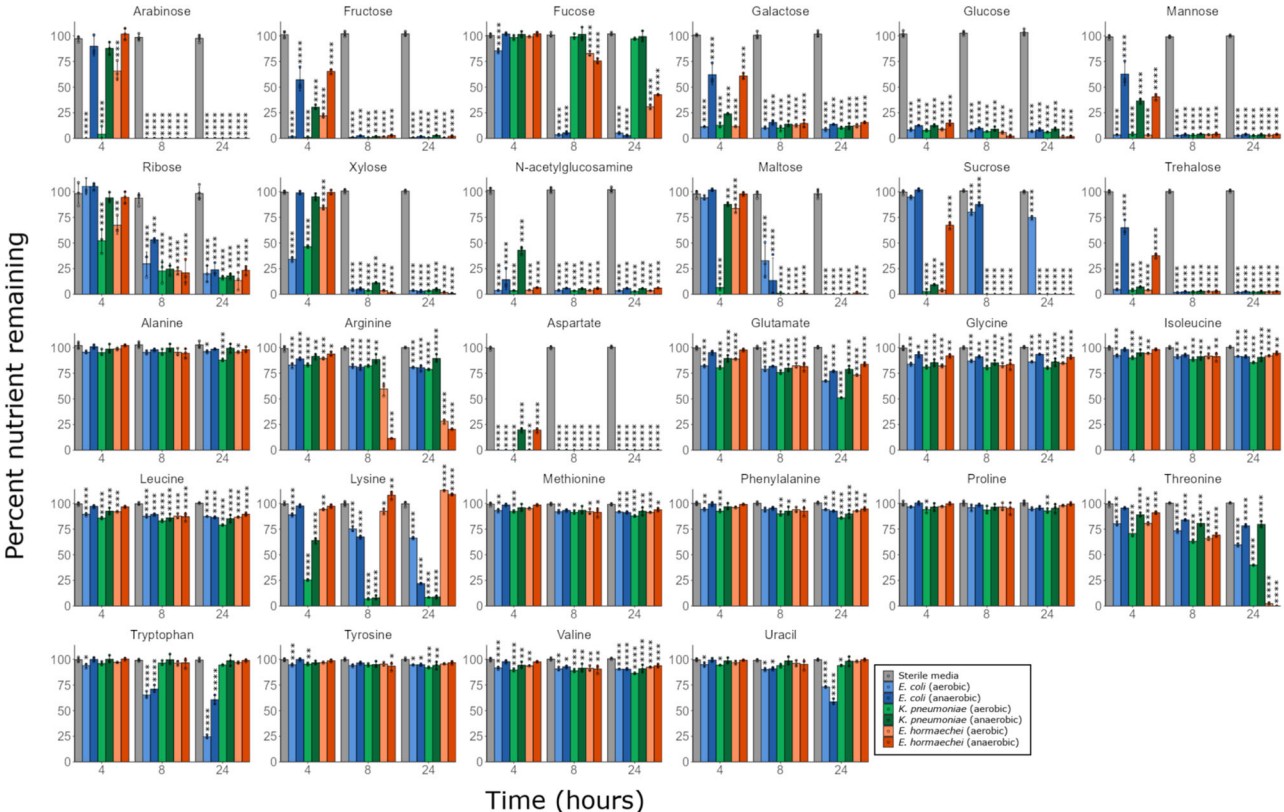

**Fig. 6 | CRE isolates had an ordered preference of nutrients when cultured in a mixture of nutrients that were elevated in antibiotic-treated faecal microbiota.** Percent of nutrients remaining in M9 minimal medium supplemented with a mixture of nutrients, inoculated with *E. coli* ST617, *K. pneumoniae* ST1026 or *E. hormaechei* ST278 and incubated under anaerobic or aerobic conditions. Nutrient concentrations were measured by $^1$H-NMR spectroscopy. Two-way mixed ANOVA followed by pairwise comparisons with Bonferroni correction. * = $P \leq 0.05$, ** = $P \leq 0.01$, *** = $P \leq 0.001$, **** = $P \leq 0.0001$, $n$ = three replicates for each CRE isolate and $n$ = 4 replicates for the sterile media controls. Data are presented as mean values ± SD. Source data and $P$ values are provided in a Source Data file.

several carbon and nitrogen sources that are available to support CRE growth following antibiotic treatment, and it is not clear whether CRE have preferences for some carbon or nitrogen sources over others. It is also not clear whether the presence of some carbon or nitrogen sources will influence the utilisation of other carbon or nitrogen sources, as has been previously demonstrated for commensal *E. coli*[23]. Therefore, the nutrient utilisation abilities of carbapenem-resistant *E. coli*, *K. pneumoniae*, and *E. hormaechei* were measured by adding a mixture of these nutrients into M9 minimal medium (lacking any other carbon or nitrogen sources) and monitoring changes in nutrient concentration over time by $^1$H-NMR spectroscopy.

*E. coli*, *K. pneumoniae* and *E. hormaechei* were able to utilise nearly all the monosaccharides and disaccharides tested (Fig. 6). When grown anaerobically, *E. coli* and *E. hormaechei* had high utilisation levels of all monosaccharides and disaccharides. *K. pneumoniae* had high utilisation of all monosaccharides and disaccharides except for fucose, which was not utilised. When grown aerobically, *E. coli* had high utilisation of all monosaccharides and disaccharides except for sucrose, which had moderate utilisation. *K. pneumoniae* had high utilisation of all monosaccharides and disaccharides except for fucose, which was not utilised. *E. hormaechei* had high utilisation of all monosaccharides and disaccharides.

*E. coli*, *K. pneumoniae*, and *E. hormaechei* showed variable amino acid utilisation patterns. Under anaerobic conditions, *E. coli* had high utilisation of aspartate and lysine, moderate utilisation of tryptophan, and low utilisation of the other amino acids (except for alanine or proline, which were not utilised). *K. pneumoniae* had high utilisation of aspartate and lysine, and low utilisation of the other amino acids (except for alanine, proline, and tryptophan, which were not utilised).

*E. hormaechei* had high utilisation of aspartate, threonine, and arginine, and low utilisation of the other amino acids (except for alanine, proline, tryptophan, and tyrosine, which were not utilised). Under aerobic conditions, *E. coli* had high utilisation of aspartate and tryptophan, moderate utilisation of glutamate, lysine, and threonine, and low utilisation of the other amino acids (except for proline, which was not utilised). *K. pneumoniae* had high utilisation of aspartate, lysine, and threonine, moderate utilisation of glutamate, and low utilisation of the other amino acids (except for tryptophan, which was not utilised). *E. hormaechei* had high utilisation of aspartate, threonine, and arginine, moderate utilisation of glutamate, and low utilisation of the other amino acids (except for proline, tryptophan, and tyrosine, which were not utilised). *E. hormaechei* produced low levels of lysine both anaerobically and aerobically.

Uracil utilisation also varied by isolate under anaerobic and aerobic conditions. *E. coli* had moderate utilisation of uracil under anaerobic and aerobic conditions. *K. pneumoniae* did not use uracil under anaerobic conditions but utilised uracil at low levels under aerobic conditions. *E. hormaechei* did not use uracil under anaerobic and aerobic conditions.

The rate at which nutrients were utilised was different for *E. coli*, *K. pneumoniae*, and *E. hormaechei* (Supplementary Fig. S17). Some nutrients were used more quickly than others, indicating that each CRE isolate had its own ordered preference of nutrients. Moreover, aerobically incubated CRE cultures utilised most nutrients more rapidly than anaerobically incubated CRE cultures.

Succinate was produced by all CRE isolates under both anaerobic and aerobic conditions. Other metabolites produced by the CRE isolates included acetate, ethanol, and formate (Supplementary Fig. S18).

In summary, these results led us to conclude that these carbapenem-resistant *E. coli*, *K. pneumoniae*, and *E. hormaechei* strains had an ordered preference for nutrients and that nutrient utilisation was different in the presence and absence of oxygen.

### Carbapenem-resistant *E. coli* utilised nutrients that were elevated in the faecal microbiota from TZP-treated mice

Next, we used a mouse model of carbapenem-resistant *E. coli* intestinal colonisation to measure changes in nutrient availability in mouse faeces following antibiotic treatment and carbapenem-resistant *E. coli* colonisation. Again, we chose to test TZP in this experiment because TZP has previously been demonstrated to promote the intestinal colonisation of CRE in humans and mice[5,19,20]. Moreover, changes in bacterial families, nutrient availability, and microbial metabolites in TZP-treated faecal microbiota were representative of changes in faecal microbiota treated with the other antibiotics that were tested in this study (Fig. 1).

Mice were administered TZP or saline for 5 days, and carbapenem-resistant *E. coli* was orally administered to the mice on the third day of TZP/saline administration (Fig. 7a). TZP-treated mice had significantly higher *E. coli* counts in their faeces compared to saline-treated mice, demonstrating that antibiotics disrupted colonisation resistance against carbapenem-resistant *E. coli* (Fig. 7b).

Faecal samples were collected from mice before TZP administration, after TZP administration but before *E. coli* administration, and after TZP and *E. coli* administration. Changes in nutrients and metabolites were measured by ¹H-NMR spectroscopy. The faeces of TZP-treated mice had significantly increased concentrations of nutrients compared to faeces collected prior to TZP treatment, including increases in monosaccharides (arabinose, fructose, galactose, glucose, xylose, N-acetylglucosamine), disaccharides (sucrose, trehalose) and amino acids (arginine, aspartate, glutamate, isoleucine, leucine, lysine, phenylalanine, proline, threonine, tryptophan, tyrosine, valine) (Fig. 7c, heatmap on the left). TZP treatment also decreased the concentration of microbial metabolites, including acetate, propionate, isobutyrate, ethanol, and lactate. Following colonisation with carbapenem-resistant *E. coli* there was a significant decrease in nutrients that we demonstrated can be utilised by *E. coli* to support its

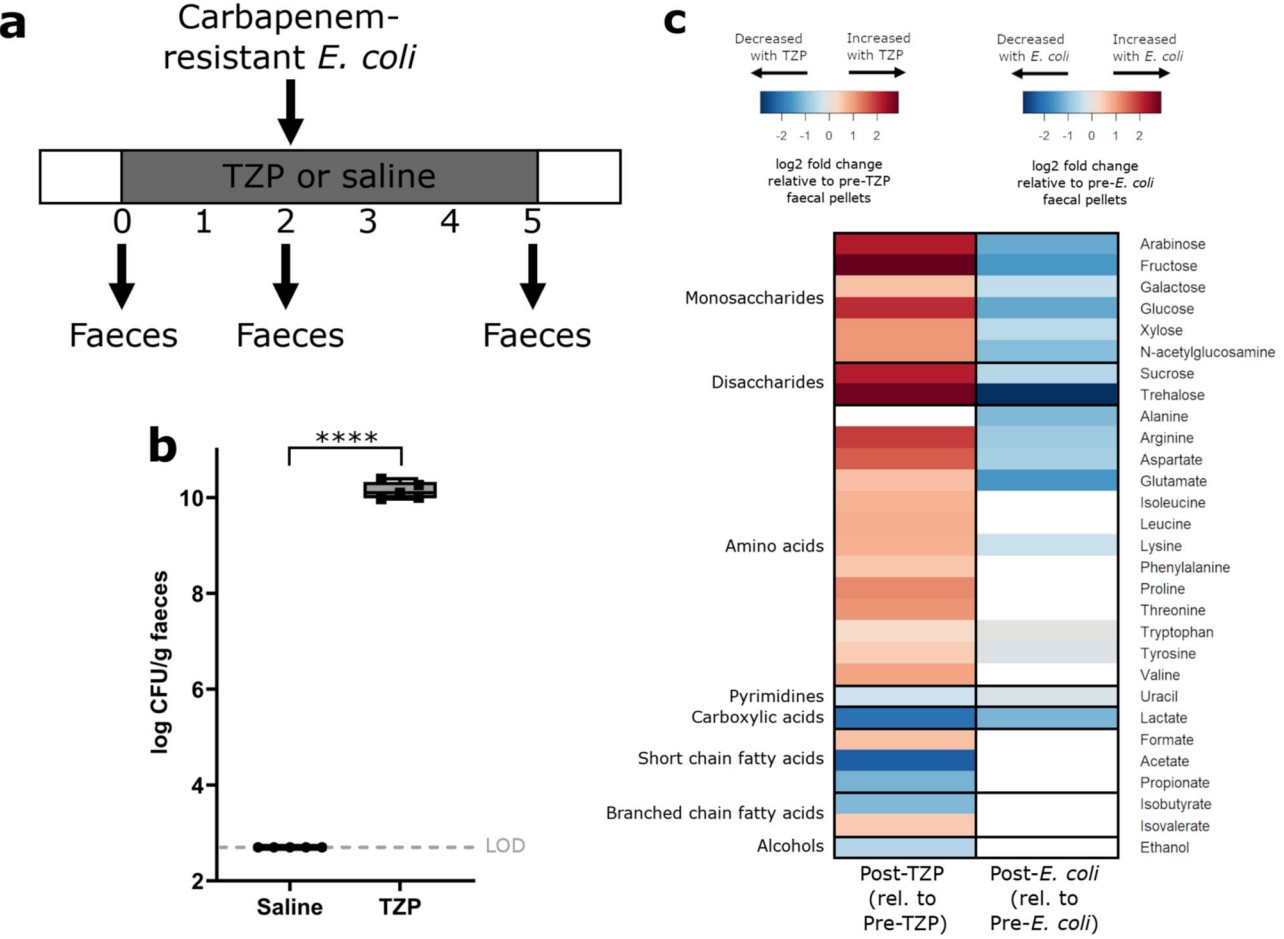

**Fig. 7 | Nutrients that were elevated with piperacillin/tazobactam (TZP) treatment decreased with carbapenem-resistant *E. coli* colonisation in faeces from a mouse model of intestinal colonisation. a** Experimental design. Mice were administered TZP or saline for 5 days and were fed by oral gavage with *E. coli* ST617 on the third day of TZP/saline administration. Faeces were collected before TZP administration, after TZP administration but before *E. coli* administration, and after TZP and *E. coli* administration. **b** *E. coli* ST617 counts from mouse faecal pellets following *E. coli* administration (at the end of the TZP dosing period). The box indicates the median as a line in the box and excludes the upper and lower 25% (quartiles) of data, and the whiskers extend to the maximum and minimum values. Unpaired *t*-test (two-sided) of log10-transformed counts. *n* = 5 mice per group, *P* ≤ 0.0001 (****). **c** Log2 fold change in nutrients and metabolites significantly

changed with interventions. For comparison of post-TZP treated faeces relative to pre-TZP treated faeces (heatmap on the left), nutrients and metabolites that were significantly increased with TZP are shown in red, and nutrients and metabolites that were significantly decreased with TZP are shown in blue (*n* = 9 mice per group from one independent experiment). For comparison of TZP-treated faeces post-*E. coli* colonisation relative to TZP-treated faeces pre-*E. coli* colonisation (heatmap on the right), nutrients and metabolites that were significantly decreased with *E. coli* colonisation are shown in blue (*n* = 4 mice per group from one independent experiment). Paired *t*-test (two-sided) with Benjamini Hochberg FDR, *p* < 0.05. Nutrients and metabolites that were not significantly different were not plotted (white). LOD Limit of detection. Source data are provided as a Source Data file.

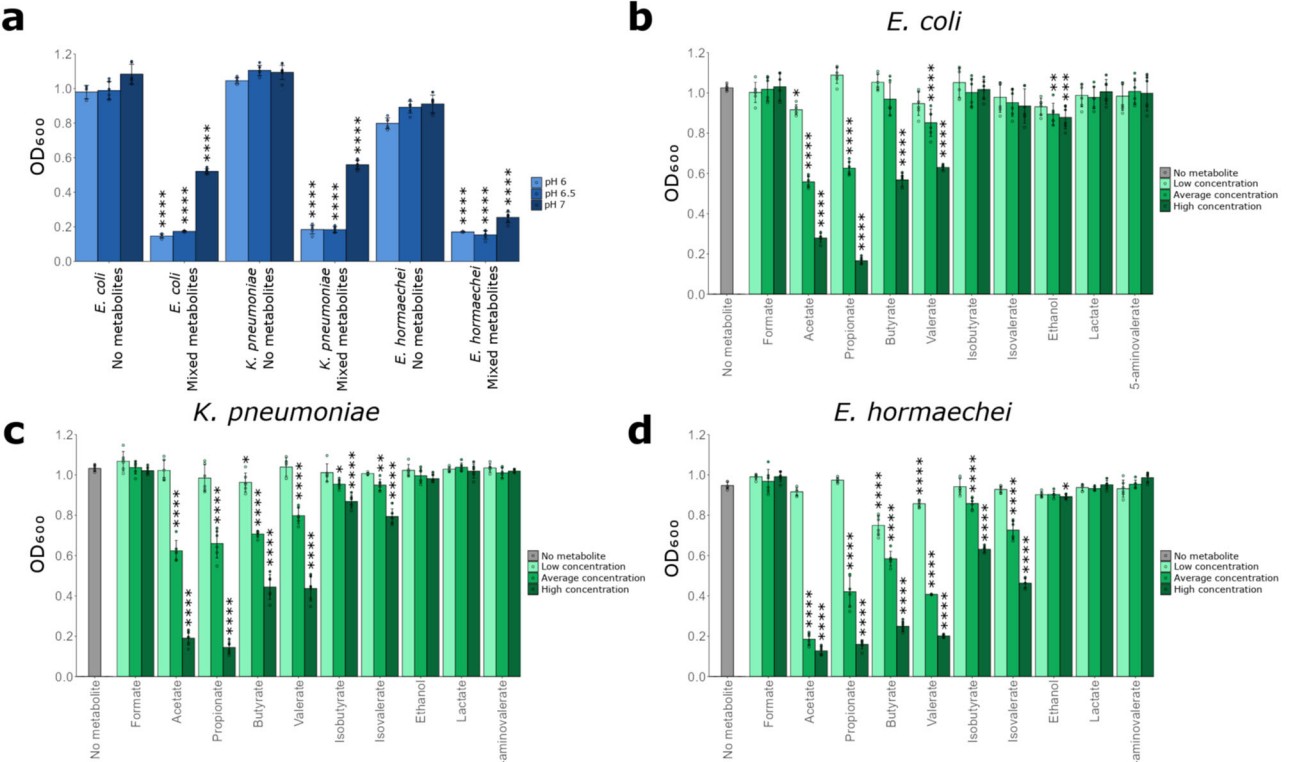

**Fig. 8 | CRE growth was inhibited by microbial metabolites that were decreased in antibiotic-treated faecal microbiota. a** CRE growth was inhibited by a mixture of metabolites that were decreased in antibiotic-treated faecal microbiota. *E. coli* ST617, *K. pneumoniae* ST1026, or *E. hormaechei* ST278 were grown in LB supplemented with a metabolite mixture (mimicking average human faecal concentrations) or unsupplemented (no metabolite control) at pH 6, 6.5 or 7. Growth of each isolate was measured with six replicates in two independent experiments. Unpaired *t*-test (two-sided) comparing metabolite mixture to the no metabolite control at the corresponding pH. **b**–**d** CRE growth was inhibited by individual metabolites in a

dose-dependent manner. Growth of *E. coli* ST617, *K. pneumoniae* ST1026 or *E. hormaechei* ST278 in LB supplemented with an individual metabolite at low, average or high concentrations (mimicking human faecal concentrations) or unsupplemented (no metabolite control) at pH 6.5. Growth of each isolate was measured with six replicates in two independent experiments. One-way ANOVA with Dunn's multiple comparison test (each metabolite was compared to the no metabolite control). * = $P \le 0.05$, ** = $P \le 0.01$, *** = $P \le 0.001$, **** = $P \le 0.0001$. Data are presented as mean values ± SD. Optical density at 600 nm, $OD_{600}$. Source data and *P* values are provided in a Source Data file.

growth, including monosaccharides (arabinose, fructose, galactose, glucose, xylose, N-acetylglucosamine), disaccharides (sucrose, trehalose), amino acids (alanine, arginine, aspartate, glutamate, lysine, tryptophan, tyrosine) and uracil (Fig. 7c, heatmap on the right).

Overall, antibiotic treatment of faecal microbiota from mice resulted in an increase in the concentration of nutrients and a decrease in the concentration of microbial metabolites, supporting observations from our ex vivo human faecal culture experiments (Fig. 1). Colonisation of the mouse intestine with carbapenem-resistant *E. coli* resulted in a decrease in nutrients that we demonstrated can be utilised by *E. coli* in vitro (Figs. 4–6), supporting our hypothesis that *E. coli* consumes these nutrients to support its growth.

### CRE growth was inhibited by microbial metabolites that were depleted in antibiotic-treated faecal microbiota

Microbial metabolites have previously been demonstrated to inhibit the growth of intestinal pathogens[7,16]. Nutrients are metabolised to produce metabolites, and therefore increased nutrient availability is linked to reduced metabolite production. We demonstrated that microbial metabolites were decreased in antibiotic-treated faecal microbiota (Fig. 1b). Therefore, we next tested whether microbial metabolites (that were decreased with antibiotics) were capable of inhibiting CRE growth.

First, we measured the concentration of microbial metabolites in faeces from 12 healthy human donors to determine the concentration of these metabolites in antibiotic-naïve faecal microbiota (Table S1). We used these measurements to define a minimum, average, and

maximum concentration to test in the metabolite inhibition assays (Table S2).

First, we tested whether a mixture of metabolites that were decreased with antibiotics (formate, acetate, propionate, butyrate, valerate, isobutyrate, isovalerate, lactate, 5-aminovalerate, and ethanol) could directly inhibit CRE growth in an inhibition assay. LB was supplemented with the metabolite mixture, where each metabolite was added to mimic the average concentration that was measured in healthy human faeces. As pH has been demonstrated to influence the ability of SCFAs to inhibit pathogen growth, the metabolite mixture was tested at three pH conditions mimicking the pH found in the healthy large intestine: 6, 6.5 and 7[16,24]. We showed that the metabolite mixture inhibited the growth of *E. coli*, *K. pneumoniae* and *E. hormaechei* at pH 6, 6.5 and 7 (Fig. 8a). However, inhibition was higher at pH 6 and 6.5 compared to pH 7. We repeated this experiment by supplementing the metabolite mixture into M9 minimal medium (at pH 6, 6.5 and 7) that was supplemented with a mixture of the nutrients that were elevated in antibiotic-treated faecal microbiota (monosaccharides, disaccharides, amino acids, uracil, and succinate). We showed that the metabolite mixture also inhibited *E. coli* and *E. hormaechei* at pH 6, 6.5 and 7, and inhibited *K. pneumoniae* at pH 6 and 6.5 (Supplementary Fig. S19).

Next, we measured CRE growth in the presence of each individual metabolite at three different concentrations: the minimum, average, and maximum concentrations measured in healthy faeces. These experiments were performed at pH 6.5 as this pH was representative of the large intestine and showed strong inhibition in the mixed

metabolite experiment (as shown in Fig. 8a). *E. coli* was inhibited by acetate at low, average, and high concentrations, by propionate, valerate, and ethanol at average and high concentrations, and by butyrate at high concentrations (Fig. 8b). *K. pneumoniae* was inhibited by butyrate at low, average, and high concentrations, and by acetate, propionate, valerate, isobutyrate, and isovalerate at average and high concentrations (Fig. 8c). *E. hormaechei* was inhibited by butyrate and valerate at low, average, and high concentrations, by acetate, propionate, isobutyrate, and isovalerate at average and high concentrations and by ethanol at the high concentration (Fig. 8d).

In summary, we demonstrated that specific microbial metabolites (that were decreased following antibiotic treatment of faecal microbiota) were inhibitory towards these carbapenem-resistant *E. coli*, *K. pneumoniae* and *E. hormaechei* strains. This led us to conclude that the significant decrease of these metabolites following antibiotics contributes to the loss of colonisation resistance against CRE.

### Carbapenem-resistant *E. coli* growth was inhibited by a mixture of microbial metabolites in a mouse model of CRE intestinal colonisation

Next, we administered a mixture of inhibitory metabolites as an intervention in a mouse model of CRE intestinal colonisation to determine whether these metabolites could inhibit the growth of carbapenem-resistant *E. coli* in vivo. TZP-treated mice colonised with carbapenem-resistant *E. coli* were administered a mixture of metabolites (acetate, propionate, butyrate, and valerate) or PBS (as a vehicle control; Fig. 9a). We chose to test these metabolites in mice as they were inhibitory against all three CRE isolates when tested individually (Fig. 8) and when tested together as a mixture (Supplementary Fig. S20). Each metabolite was administered to the mice in their

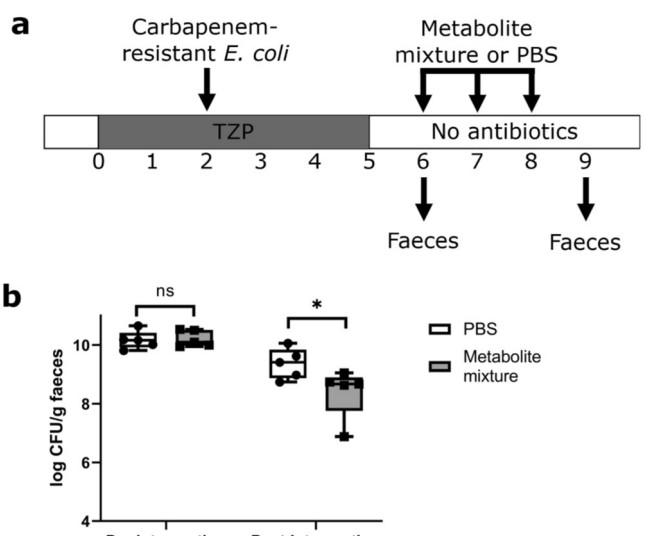

**Fig. 9 | Microbial metabolites significantly decreased carbapenem-resistant *E. coli* growth in faeces from a mouse model of intestinal CRE colonisation.**
**a** Experimental design. Mice were administered TZP for 5 days and were orally administered *E. coli* ST617 on the third day of TZP administration. Mice were then fed by oral gavage with a mixture of metabolites (glycerol triacetate, glycerol tripropionate, glycerol tributyrate, and glycerol trivalerate) or phosphate buffered saline (PBS) for 3 days. **b** *E. coli* ST617 counts were quantified from mouse faecal pellets just before metabolite mixture or PBS administration (pre-intervention) and at the end of metabolite mixture or PBS administration (post-intervention). The box indicates the median as a line in the box and excludes the upper and lower 25% (quartiles) of data, and the whiskers extend to the maximum and minimum values. Mann–Whitney U test (two-sided) of log10-transformed counts. $n = 5$ mice per group from one independent experiment, $P > 0.9999$ for pre-intervention (ns), $P = 0.0397$ for post-intervention (*). Source data are provided as a Source Data file.

triglyceride form (i.e. as a mixture of glycerol triacetate, glycerol tripropionate, glycerol tributyrate, and glycerol trivalerate) to avoid rapid absorption of SCFAs following oral gavage and to obtain sufficient concentrations of these SCFAs in the gastrointestinal tract[7]. The triglyceride forms of these SCFAs are hydrolysed by lipases to release acetate, propionate, butyrate, and valerate, respectively, in the intestine[25].

Prior to the administration of the metabolite mixture or PBS to mice, there was no significant difference in *E. coli* counts in the faeces from each group. However, there was a significant decrease in carbapenem-resistant *E. coli* counts in the faeces of metabolite-treated mice compared to PBS-treated mice (Fig. 9b). After three doses of the metabolite mixture or PBS, metabolite-treated mice had an average of 87% less *E. coli* CFU per gram of faeces compared to PBS-treated mice.

Overall, we demonstrated that metabolites that were decreased in antibiotic-treated human faecal microbiota (acetate, propionate, butyrate and valerate) were able to inhibit carbapenem-resistant *E. coli* in a mouse model of intestinal CRE colonisation, highlighting their important role in providing colonisation resistance against CRE.

## Discussion

In this study we demonstrated that antibiotic-mediated disruption of the gut microbiota (using antibiotics that promote CRE intestinal colonisation) led to the development of a nutrient-enriched and metabolite-depleted niche that promoted CRE growth. We showed that CRE can use these elevated nutrients as carbon and nitrogen sources to support their growth, where CRE show an order of preference for specific nutrients. CRE growth was higher on these nutrients in an oxygenated environment, which has been associated with antibiotic treatment. We also showed that specific microbial metabolites that were decreased with antibiotics were able to inhibit CRE growth.

In this study we tested antibiotics that are frequently used clinically and known to promote the intestinal colonisation with CRE: carbapenems (MEM, IPM, ETP), penicillin/β-lactamase inhibitor (TZP), fluoroquinolones (CIP) and cephalosporins (CRO, CAZ, CTX)[5]. Although we have not tested all possible antibiotics that may promote the intestinal colonisation with CRE, these were a feasible number of antibiotics to test and logical choices to prioritise for this study. Faecal cultures treated with these eight antibiotics caused broadly consistent enrichments of nutrients and depletions of metabolites. Future studies could explore the impact of other antibiotics on nutrient enrichment, metabolite depletion, and CRE growth.

We showed that healthy faecal microbiota treated with antibiotics that promote CRE intestinal colonisation resulted in a decrease of several bacterial families, in particular *Bifidobacteriaceae* (decreased with all antibiotics tested), families belonging to *Bacteroidales* (decreased with carbapenems, TZP, and CIP), and *Coriobacteriaceae* (decreased with carbapenems). Our results are supported by previous studies in healthy humans which also demonstrated a decrease in *Bifidobacterium* and *Bacteroides* with these antibiotics[26–31]. Our results are also supported by a recent study which demonstrated that CRE-positive patients had lower amounts of *Bacteroides dorei*, *Bifidobacterium bifidum*, *Bifidobacterium pseudocatenulatum*, and *Collinsella aerofaciens* in their faeces compared to CRE-negative patients[32]. Moreover, a recent study demonstrated that *Bacteroidota* and *Actinobacteria* (including *Bifidobacterium*) engraft very highly within the intestine following faecal microbiota transplantation[33]. This highlights the possible role of *Bifidobacterium* and *Bacteroides* in decolonising CRE-positive patients following faecal microbiota transplantation.

Polysaccharides, proteins, and mucins are major food sources for the gut microbiota in the large intestine. Primary degraders break down these complex substrates, resulting in the extracellular release of breakdown products (oligosaccharides, disaccharides, monosaccharides, peptides, and amino acids)[15]. These simple compounds

can support the growth of secondary degraders. *Bacteroides* are important primary degraders (but can also degrade simple compounds) and *Bifidobacterium* are important secondary degraders in the gut[34]. Therefore, this supports the observation that decreases in the abundance of these bacteria following antibiotics results in an enrichment of simple nutrients.

Members of the *Bifidobacteriaceae*, *Bacteroidales*, and *Coriobacteriaceae* have been shown to utilise many of the nutrients that were elevated with antibiotics. *Bifidobacterium* have been shown to utilise most of the monosaccharides and disaccharides tested in this study[35]. Amino acid utilisation has not been well characterised in *Bifidobacterium*, however some *Bifidobacterium* species have been shown to utilise alanine, aspartate, glutamate, and threonine as sole carbon sources[36,37]. *Bacteroides* have been shown to utilise all the monosaccharides tested in this study and can utilise amino acids[13,38,39]. *Collinsella* can utilise most monosaccharides and disaccharides tested in this study and are known to metabolise amino acids[40,41].

The nutrient-niche hypothesis predicts that expansion of *Enterobacteriaceae* in an antibiotic-disrupted gut microbiome is due to an increase in the availability of nutrients, where competition for these nutrients would normally restrict *Enterobacteriaceae* growth[14]. We demonstrated that antibiotics (that promote susceptibility to CRE) increased the availability of nutrients that can support CRE expansion within the gut. The monosaccharides, disaccharides, and amino acids of interest in this study were derived from the breakdown of complex substrates, including starch (glucose, maltose, trehalose)[42–45], inulin (fructose, glucose, sucrose)[46], pectin (glucose, arabinose, galactose, xylose, mannose, fucose)[47], xylan (xylose, arabinose)[48], arabinogalactan (arabinose, galactose)[49], mucin (N-acetylglucosamine, galactose, fucose, amino acids)[50], peptone (amino acids)[51,52], and yeast extract (amino acids, ribose, uracil)[53]. Our results led us to conclude that CRE have a growth advantage in antibiotic-perturbed gut microbiota by acting as secondary fermenters of monosaccharides, disaccharides, and amino acids, highlighting the important role of microbial cross-feeding in antibiotic-disrupted gut microbiota. We also showed that nutrient utilisation profiles could vary between different strains of the same carbapenem-resistant species. This highlights the need to further compare the nutrient utilisation abilities amongst different CRE isolates to determine whether expanded nutrient utilisation abilities may drive the global spread of prevalent clonal groups.

We also demonstrated that nutrient utilisation was influenced by the presence of other nutrients. Other studies have also shown that the presence of some nutrients influences the utilisation of others. Fucose has been shown to stimulate the utilisation of ribose by commensal *E. coli*[23]. Maltose utilisation gene expression was elevated in commensal *E. coli* grown on mucus[11]. Therefore, these results highlight the importance of studying nutrient utilisation as a mixture of nutrients in addition to their utilisation as sole nutrient sources.

Previous studies demonstrated that antibiotics could enrich nutrients to support the growth of intestinal pathogens. Cefoperazone (which promoted *Clostridioides difficile* intestinal colonisation) increased nutrients (mannitol, fructose, sorbitol, raffinose, stachyose, and some amino acids) and decreased metabolites (valerate and isovalerate) in the caecal microbiota of mice[54]. In vitro experiments demonstrated that these nutrients supported *C. difficile* growth. Another study demonstrated that sialic acid was increased in the caecal microbiota of mice treated with antibiotics that promote susceptibility to *Salmonella enterica* serovar Typhimurium or *C. difficile*[10]. *S.* Typhimurium and *C. difficile* mutants that were unable to utilise sialic acid showed impaired growth within the intestine. In this study we found an enrichment of nutrients associated with polysaccharides, mucin, and proteins, indicating that antibiotics cause a disruption of secondary fermentation in the gut, where enrichment of these simple nutrients can support CRE growth.

A limitation of this study is that faecal microbiota were only analysed for nutrients and metabolites using ¹H-NMR spectroscopy. Although this is an untargeted metabolic profiling technique that can measure a wide variety of relevant nutrients and metabolites of interest, it is not an exhaustive technique and cannot profile all nutrients and metabolites in our samples. There may be changes in other nutrients and metabolites that we did not measure (e.g. decreased primary fermentation of polysaccharides, mucin, or proteins with antibiotics). Therefore, if the utilisation of complex substrates by primary fermenters (such as *Bacteroides*) decreases, these substrates may provide an additional nutrient source for CRE. This potential role for primary fermenters also highlights the importance of continued investigation of the role of other nutrients and metabolites in promoting or inhibiting CRE intestinal colonisation and expansion.

Oxygen is a limiting resource for *Enterobacteriaceae* in the gut, and oxygenation of the gut that occurs following antibiotics has been hypothesised to drive the expansion of *Enterobacteriaceae*[14]. Our results demonstrate that an increase in oxygen availability results in an increase in CRE growth on nutrients that were enriched with antibiotics (compared to growth under anaerobic conditions). This observation is supported by a recent longitudinal shotgun metagenomics sequencing analysis of stool from CRE-positive patients which showed that aerobic respiration and oxidative phosphorylation pathways (e.g. the pentose phosphate pathway) were enriched during CRE colonisation (where antibiotic usage was common)[32]. Rivera-Chávez and colleagues demonstrated that antibiotic treatment led to increased gut oxygenation that resulted in the aerobic expansion of *S.* Typhimurium, demonstrating the advantage that oxygen can provide to facultatively anaerobic pathogens[55]. Overall, our work supports the hypothesis that CRE growth is enhanced on carbon and nitrogen sources (that become available due to antibiotic treatment) in the presence of oxygen.

The metabolism of monosaccharides and disaccharides produces SCFAs and the metabolism of amino acids produces SCFAs and BCFAs[15,56]. We demonstrated that antibiotic treatment of the faecal microbiota results in the disruption of nutrient metabolism, leading to the development of a metabolite depleted environment. We showed that formate, acetate, propionate, butyrate, valerate, isobutyrate, isovalerate, lactate, 5-aminovalerate, and ethanol were decreased by antibiotics. *Bifidobacterium*, *Bacteroides* and *Collinsella* (taxa that were decreased with antibiotics) are known to produce many of these metabolites. *Bifidobacterium* produces acetate, lactate, ethanol, and formate[15,57]. *Bacteroides* produces acetate, propionate, lactate, formate, ethanol, isobutyrate, and isovalerate[15,58]. *Collinsella* produces ethanol, formate and lactate[40].

Sorbara and colleagues showed that acetate, propionate, and butyrate were decreased in the caecal contents from mice treated with ampicillin, and these metabolites inhibited antibiotic-resistant *K. pneumoniae* and *E. coli* growth under pH conditions found in the caecal contents from antibiotic-naïve mice (pH 5.75–6.25)[16]. Djukovic et al. also showed that butyrate inhibited multidrug-resistant *K. pneumoniae* growth in vitro, and showed a negative correlation between the levels of butyrate and the levels of multidrug-resistant *Enterobacteriaceae* in human faeces[17]. Our work adds novel insights to these findings, as we demonstrated that broad-spectrum antibiotics (that promote CRE intestinal colonisation in humans) depleted a wide range of microbial metabolites and generated a metabolite-depleted niche that no longer inhibited CRE growth. We showed that ten microbial metabolites were decreased with antibiotics (formate, acetate, propionate, butyrate, valerate, isobutyrate, isovalerate, ethanol, lactate, 5-aminovalerate). We showed that a mixture of these ten metabolites was inhibitory against CRE growth at pH values representative of the human large intestine (pH 6, 6.5 and 7). However, the metabolite mixture was more inhibitory at pH 6 and 6.5 compared to pH 7. Although Sorbara and

colleagues did not show that a mixture of acetate, propionate, and butyrate inhibited *K. pneumoniae* or *E. coli* growth at pH 7, it is possible that our metabolite mixture showed inhibition at pH 7 due to the inclusion of additional inhibitory metabolites in our metabolite mixture. This is supported by previous observations that valerate can inhibit the growth of *C. difficile* at pH 7[7]. We further demonstrated that seven of these ten metabolites were capable of inhibiting CRE growth (acetate, propionate, butyrate, valerate, isobutyrate, isovalerate, and ethanol).

We used faecal samples from 12 healthy human donors to quantify the average, minimum, and maximum metabolite concentrations to test in the experiments outlined in Fig. 8. Although this is not a large cohort of healthy human donors, the metabolite concentrations measured from faeces in our study were comparable to concentrations measured in other studies (where sample sizes ranged from 5 to 93 faecal donors)[59,60]. Future studies should measure the concentration of these ten metabolites in a larger cohort of healthy human donors to confirm the metabolite concentrations found in healthy human faeces.

There have been some investigations into the mechanisms that acetate, propionate, and butyrate use to inhibit bacterial growth. A previous study by Sorbara et al. demonstrated that acetate, propionate, and butyrate inhibited *E. coli* and *K. pneumoniae* growth through intracellular acidification[16]. They showed that these three SCFAs (present in their nonionised form at low pH) diffused across the bacterial membrane into the cytoplasm. Once inside the bacterial cell these SCFAs dissociated into their ionised forms, releasing protons into the cytoplasm and acidifying the intracellular pH. In another study Park and colleagues demonstrated that butyrate exhibited strain-dependent inhibitory activity against *Bacteroidales*, which was impacted by the utilisation of distinct sugars in a context-dependent manner[61]. They also demonstrated that variation in Acyl-CoA thioesterase and transferase activity governed differences to butyrate resistance in *Bacteroides*. However, in this study we demonstrated that additional metabolites (valerate, isobutyrate, isovalerate, ethanol) also caused growth inhibition of CRE, and further research is required to solve the mechanisms that these inhibitory metabolites use to restrict intestinal CRE colonisation. Moreover, it is unclear whether these seven inhibitory metabolites may act using one mechanism of inhibition or whether they may mediate inhibition by multiple different mechanisms. We saw differences in the metabolites that inhibited the growth of the *E. coli*, *K. pneumoniae*, and *E. hormaechei* strains tested, and it is not clear whether there are differences in the mechanisms in these three different species. It is also unclear whether these metabolites may have synergistic inhibitory effects, or whether these mechanisms may be impacted by different gut environmental conditions. Future studies are needed to investigate how these metabolites impact pathogens, other members of the gut microbiota, and the host to provide colonisation resistance against CRE.

In this study we investigated the impact of microbial metabolites on the growth of CRE isolates. However, it is important to further investigate the impact of microbial metabolites on the host, as this may also impact the development of invasive CRE infections. For example, both butyrate and valerate have been demonstrated to improve gut barrier integrity[62,63]. Therefore, a depletion of these metabolites following antibiotic treatment may promote the translocation of CRE from the gut to the bloodstream.

Further development of our metabolite mixture into a treatment for CRE intestinal colonisation will require additional optimisation. Future studies should determine the optimal concentrations of metabolites, optimal dosing frequency, and optimal dosing duration to best inhibit CRE growth. These experiments should also follow changes in CRE intestinal colonisation in response to the metabolite mixture over time. Finally, antibiotic treatment is known to increase the pH of the gut[16]. Therefore, acidification of the gastrointestinal tract

(e.g. via colonisation with gut commensals that can ferment substrates) may further improve the efficacy of the metabolite mixture to inhibit CRE growth. Future studies should investigate the synergistic effects of inhibitory gut commensals and microbial metabolites.

We did not measure the concentration of the supplemented metabolites in the mouse faeces or intestinal content after the metabolite mixture was administered to mice. However, SCFAs are routinely administered as their triglyceride versions as it is well established that these compounds are effective at increasing the luminal concentrations of the SCFAs in the distal gut[64]. Previous studies that administered glycerol tributyrate (aka tributyrin) to antibiotic-treated mice showed a significant increase in butyrate concentration in the colon contents and caecal contents of mice, where CRE are known to colonise[64,65]. Future studies should quantify metabolite concentrations in mouse faeces and in different segments of the gastrointestinal tract following administration of the metabolite mixture to better define the pharmacokinetics of this potential new therapeutic.

There were some additional limitations associated with the mouse experiments outlined in this study. Firstly, it is important to remember that mice are not a perfect model for humans for microbiome studies. There are differences in the microbiome taxonomic compositions and diets of humans and mice, among other factors[66]. Although we do not see the exact same changes in nutrients and metabolites in human and mouse faeces in response to TZP, our data demonstrates the same overall trends with TZP treatment: an increase in monosaccharides, disaccharides, and amino acids, and a decrease in SCFAs, BCFAs, ethanol, and lactate. Next, in this study mice were housed five per cage rather than individually. As mice engage in coprophagia, it is possible that mice were reinoculated with carbapenem-resistant *E. coli* that were ingested from faeces[67]. This may be improved by housing mice individually, however mice also ingest their own faeces and so a certain amount of coprophagy is difficult to avoid in mouse experiments. As the mice caged together are from the same treatment group, faeces from all mice in the same cage are likely similar. Therefore, we believe ingested faeces would be comparable in individually housed mice and group housed mice. Finally, a limitation of the mouse experiments is that they were performed as one independent experiment. However, our mouse experiments were designed using power calculations to achieve the desired statistical power. Designing an experiment to achieve a desired power (e.g. $\alpha = 0.05$ and a power of 0.80) and repeating this same experimental design several times may result in a higher power than is needed and overpower the study, resulting in the use of more animals than is needed[68]. In accordance with the 3Rs principle of reduction, we minimised animal use to obtain sufficient data to answer our research question[69]. Our in vivo results are not a stand-alone data set, as they are substantiated by complementary data from our ex vivo and in vitro experiments that provide strong support for our in vivo data. Future studies that continue the development of this metabolite mixture as a new treatment could repeat this experiment while also generating new data to make the most of using these animals for experimentation (e.g. testing multiple doses of the metabolite mixture)[64,65].

In summary, we provide evidence that antibiotics create an intestinal niche that supports CRE expansion by enriching for nutrients that support CRE growth and by depleting metabolites that are inhibitory towards CRE growth. Understanding how antibiotics disrupt colonisation resistance against CRE is critical for the rational design of microbiome therapeutics that prevent or treat CRE intestinal colonisation. These microbiome therapeutics could reduce the risk of patients developing invasive CRE infections, reduce the recurrence of invasive CRE infections in chronically colonised patients, and reduce the spread of CRE to susceptible patients. A second-generation microbiome therapeutic may be composed of gut commensals that can outcompete CRE for nutrients and convert them into inhibitory metabolites. A third-generation microbiome therapeutic may be

composed of a metabolite or mixture of metabolites that inhibit CRE growth, which could be administered to CRE colonised patients while they are receiving antibiotic treatment.

In the short term, our results could be used to help reduce the risk of patients becoming colonised with CRE. Our data could guide the prescription of antibiotics whereby clinicians avoid prescribing antibiotics that elevate specific nutrients that support CRE growth and deplete specific metabolites that inhibit CRE growth. Our data could also be used to identify patients that have an increased risk for CRE intestinal colonisation by screening faecal samples for these nutrients and metabolites.

## Methods

### Materials
Table S3 lists the reagents and oligonucleotides used in this study.

### Carbapenem-resistant *Enterobacteriaceae* isolates
*E. coli* ST617 (NDM-5), *K. pneumoniae* ST1026 (NDM-1) and *E. hormaechei* ST278 (NDM-1; part of the *Enterobacter cloacae* complex) were isolated from patients with intestinal CRE colonisation from the Imperial College Healthcare NHS Trust (London, UK) via rectal swab (with approval from the London - Queen Square Research Ethics Committee, 19/LO/0112). We also tested *E. coli* ST167 (NDM-4, NCTC 14333), *E. coli* ST410 (OXA-48, NCTC 14324), *K. pneumoniae* ST258 (KPC-3, NCTC 13438) and *K. pneumoniae* ST11 (KPC-3, NCTC 14327) as these are prevalent clonal groups that are responsible for the rapid emergence of CRE clinical isolates worldwide[70].

### Whole genome sequencing of the CRE patient isolates
Genomic DNA from the *E. coli*, *K. pneumoniae* and *E. hormaechei* patient isolates was prepared using the GenElute Bacterial Genomic DNA Kit (Sigma-Aldrich, UK), and the Gram-negative DNA extraction protocol. Genomic DNA was sent to MicrobesNG (https://microbesng.com/) for whole genome sequencing using Illumina Novaseq 6000 platform with 250 bp pair-end protocol.

### Genomic analysis to determine multi locus sequence types (MLST) and antimicrobial resistance genes of the CRE patient isolates
Raw reads were trimmed to remove sequences and low-quality bases with Trimmomatic v0.39 (https://github.com/timflutre/trimmomatic). Bacterial species from assembled genomes was confirmed using Kraken2 v2.0.8-beta and its full bacterial database (http://ccb.jhu.edu/software/kraken2/, downloaded on 26/05/2022)[71]. Draft genomes were generated de novo using SPAdes v3.15.2[72]. Assembly statistics were checked using QUAST v5.0.2[73]. Acquired antimicrobial resistance genes were detected from draft genomes using ABRicate v1.0.1 (https://github.com/tseemann/abricate) with the ResFinder database (https://cge.food.dtu.dk/services/ResFinder/)[74,75]. The acquired antimicrobial resistance genes are listed in Supplementary Data 1. Multi locus sequence types (MLST) were determined from draft genomes using MLST v2.19.0 (https://github.com/tseemann/mlst). The following MLST databases were used to determine MLST: *E. hormaechei* − *Enterobacter cloacae* MLST scheme (hosted at https://pubmlst.org, accessed on 28/04/2023), *E. coli* − Achtman's *E. coli* MLST scheme (hosted at https://pubmlst.org, accessed on 17/05/2023), *K. pneumoniae* − *Klebsiella pneumoniae* MLST scheme (hosted at https://bigsdb.pasteur.fr/klebsiella/, accessed on 28/04/2023).

### Minimum inhibitory concentrations (MICs)
MICs for MEM, IPM, ETP and TZP were measured for all CRE isolates used in this study following standard protocols (Table S4)[76]. Breakpoint values of MICs were assigned as follows according to EUCAST guidelines: meropenem ≤2 mg/L sensitive, >8 mg/L resistant; imipenem ≤2 mg/L sensitive, >4 mg/L resistant; ertapenem ≤0.5 mg/L sensitive, >0.5 mg/L resistant; piperacillin/tazobactam ≤8 mg/L sensitive, >8 mg/L resistant[77].

### Ethical approval
Ethical approval was received by the London−Queen Square Research Ethics Committee (19/LO/0112) and the South Central−Oxford C Research Ethics Committee (16/SC/0021 and 20/SC/0389) to collect faecal samples from human donors. Faecal donors were healthy human donors aged between 18−65 years old and had not received antibiotic treatment in the 6+ months prior to donation. Both male and female donors were recruited (approximately equal numbers of males and females for each experiment) as our findings apply to both sexes and there is no biological reason to favour one sex over the other. Informed consent was obtained from human participants.

### Faecal culture experiments
In our first ex vivo faecal culture experiment we measured the effects of eight broad-spectrum antibiotics on the faecal microbiota collected from 11 healthy human donors (6 males, 5 females). Female faecal donors were $33.2 \pm 16.0$ years old and male faecal donors were $29.7 \pm 6.4$ years old. Faeces from each donor was used to seed all experimental conditions, including each of the eight antibiotic-treated cultures and the antibiotic-naïve control culture (for a total of nine test conditions per donor). Faecal culture experiments were performed under anaerobic conditions using a standardised distal gut growth medium mimicking nutrients found in the distal gut, including polysaccharides, proteins, and mucin[78]. For each donor, fresh faeces were inoculated into the distal gut growth medium at a 2% (w/v) concentration. Faecal cultures were supplemented with water (antibiotic-naïve control) or one of the following antibiotics (at concentrations measured in faeces): 2 μg/ml MEM, 2 μg/ml IPM, 37 μg/ml ETP, 139 μg/ml TZP, 152 μg/ml CRO, 22 μg/ml CAZ, 2 μg/ml CTX, 139 μg/ml CIP[18,27−29,79−82]. These antibiotics are known to promote the intestinal colonisation with CRE and are used frequently in routine clinical practice[5]. Faecal cultures were run in triplicate and incubated anaerobically at 37 °C for 24 h. Samples were analysed by proton nuclear magnetic resonance (¹H-NMR) spectroscopy (for nutrient and metabolite profiling) and 16S rRNA gene sequencing with 16S rRNA gene qPCR (for bacterial composition and biomass).

A second set of ex vivo faecal culture experiments was performed to measure CRE growth in TZP-treated and antibiotic-naïve faecal microbiota. Fresh faecal samples were donated from five healthy human donors (two males, three females), and these faecal samples were separate from the faecal samples collected above. Female faecal donors were $31.3 \pm 5.5$ years old and male faecal donors were $31.0 \pm 9.9$ years old. For each donor, fresh faeces were inoculated into the distal gut growth medium supplemented with TZP (139 μg/ml) or water and spiked with $10^3$ CFU/ml of one of the following CRE isolates: *E. coli* ST617, *E. coli* ST167, *E. coli* ST410, *K. pneumoniae* ST258 or *K. pneumoniae* ST11. Faecal cultures were run in duplicate and incubated anaerobically at 37 °C for 24 h. Samples were plated on Brilliance CRE Agar (Thermo Fisher Scientific, UK) to quantify CRE growth.

### 16S rRNA gene sequencing and 16S rRNA gene qPCR
DNA was extracted from 250 μl of faecal culture sample using the DNeasy PowerLyzer PowerSoil Kit (Qiagen, Germany) according to manufacturer's instructions, with the addition of a bead beating step for 3 min at speed 8 in a Fast-Prep 24 bead beater (MP Biomedicals, USA). DNA was stored at −80 °C until it was ready to be used.

Illumina's 16S metagenomic sequencing library preparation protocol was used to generate sample libraries using primers to amplify the V1-V2 regions of the 16S rRNA gene[83]. The forward primer mix contained the 28F-YM, 28F-Borrelia, 28F-Chloroflex, and 28F-Bifdo primers mixed at a ratio of 4:1:1:1, and the reverse primer was 388R (Table S3). The SequalPrep Normalisation Plate Kit (Life Technologies,

UK) was used to clean up and normalise the index PCR reactions. The NEBNext Library Quant Kit for Illumina (New England Biolabs, UK) was used to quantify sample libraries. Sample libraries were sequenced on an Illumina MiSeq platform (Illumina Inc., UK) using the MiSeq Reagent Kit v3 (Illumina) and paired-end 300 bp chemistry. 16S rRNA gene sequencing data were imported into R and processed using the standard DADA2 pipeline (version 1.18.0)[84]. The SILVA bacterial database version 138.1 was used to classify the sequence variants (https://www.arb-silva.de/).

16S rRNA gene qPCR was performed to quantify the bacterial biomass according to a previously published protocol[7]. Each 20 μL reaction volume contained: 1x Platinum Supermix with ROX, 1.8 μmol/L BactQUANT forward primer, 1.8 μmol/L BactQUANT reverse primer, 225 nmol/L BactQUANT probe, 5 μL DNA, and PCR grade water. The BactQUANT forward primer, BactQUANT reverse primer, and BactQUANT probe sequences are listed in Table S3. A standard curve was generated using 10-fold dilutions of *E. coli* DNA. Each sample, standard, and negative control was measured in triplicate. The Applied Biosystems StepOnePlus Real-Time PCR System (Applied Biosystems, Waltham, Massachusetts) was used for amplification and real-time fluorescence detection. The PCR cycling conditions were as follows: 50 °C for 3 min, 95 °C for 10 min, and 40 cycles of 95 °C for 15 s and 60 °C for 1 min.

16S rRNA gene sequencing data was expressed as absolute abundances using the 16S rRNA gene qPCR data according to a previously published protocol as follows[7]:

$$\text{Absolute abundance of taxa} = \text{relative abundance of taxa} \times \left( \frac{\text{16S rRNA gene copy number in sample}}{\text{highest 16S rRNA gene copy number in sample set}} \right)$$

### ¹H-NMR spectroscopy
Supernatants were prepared from human faeces by vortexing 300 mg of faeces in 900 μl of high-performance liquid chromatography-grade water for 10 min at 3000 rpm followed by centrifugation at 17,000 × $g$ for 10 min at 4 °C. Supernatants were prepared from faecal culture samples by centrifugation at 17,000 × $g$ for 10 min at 4 °C. Supernatants were prepared from mouse faeces by vortexing 20 mg of faeces in 560 μl of high-performance liquid chromatography-grade water for 10 min at 3000 rpm followed by centrifugation at 17,000 × $g$ for 10 min at 4 °C.

¹H-NMR spectroscopy was performed according to a previously published protocol[7]. Sample supernatants were randomised and defrosted at room temperature for 1 h. Sample supernatants were centrifuged for 10 min at 17,000 × $g$ and 4 °C and the pellet was discarded. For human faeces and faecal culture samples, 400 μl of supernatant was added to 250 μl NMR buffer (28.85 g $Na_2HPO_4$, 5.25 g $NaH_2PO_4$, 1 mM 3-(Trimethylsilyl)propionic-2,2,3,3-d₄ sodium salt (TSP), 3 mM $NaN_3$, deuterium oxide to 1 L, pH 7.4). For mouse faecal samples 540 μl of supernatant was added to 60 μl NMR buffer (20.4 g $KH_2PO_4$, 5.8 mM TSP, 2 mM $NaN_3$, deuterium oxide to 100 mL, pH 7.4). Each 5 mm NMR tube was filled with 600 μl of sample and loaded into the SampleJet system.

¹H-NMR spectroscopy was performed using the Bruker AVANCE III HD 800 MHz spectrometer (Bruker Bio-Spin, Rheinstetten, Germany) using a 5 mm CPTCI 1H-13C/15 N/D Z-gradient cryoprobe. 1D spectra (a standard NOESYGPPR1D pulse sequence (RD-90°-t1-90°-tm-90°-ACQ)) as well as 2D spectra (standard 2D JRESGPPRQF pulse sequence) were acquired. A recycle delay of 4 s and mixing time of 100 ms was used. The 90° pulse length was ~10 μs. For the human faecal samples and faecal culture samples 32 scans were recorded. For the mouse faecal samples 128 scans were recorded.

All NMR spectra were processed in Topspin (v3.2.6) in the same manner, including an exponential window function (line broadened by 0.3 Hz), zero-order auto phasing (using command apk0), baseline correction (using command abs), and referencing with TSP (using

command sref). NMR spectra were imported into MATLAB r2019b (The Mathworks, USA) using a custom script. Spectra were aligned and the water (4.6–5 ppm) and TSP (−0.2–0.2 ppm) peaks were cut. Compounds were identified by comparison to spectral databases in Chenomx NMR Suite v9.02 (Chenomx, Canada). Representative peaks were integrated for quantification of compounds. A signal-to-noise ratio cut-off value of 3:1 was used to define the limit of detection for peak detection[85]. The minimal signal-to-noise ratio was 6.1:1 for the 32 scan human faecal dataset, 3.8:1 for the 32 scan faecal culture dataset, and 4.0:1 for the 128 scan mouse faecal dataset.

### rCCA modelling
rCCA modelling was used to correlate 16S rRNA gene sequencing data (family level) with ¹H-NMR data for faecal cultures samples using the mixOmics R package version 6.22.0, using the shrinkage method to determine the regularisation parameters[7,21]. Unit representation plots (generated using the plotIndiv function) show each faecal culture sample as a point projected into the XY-variate space. In correlation circle plots (generated using the plotVar function) variables projected in the same direction from the origin have a positive correlation, and variables projected in opposite directions from the origin have a negative correlation. Variables sitting at farther distances from the origin have stronger correlations than variables sitting closer to the origin.

### Carbon and nitrogen utilisation assays
Because CRE growth is promoted in antibiotic-treated faecal microbiota with disrupted colonisation resistance, compounds that were enriched in antibiotic-treated faecal microbiota were tested as nutrients in growth assays to determine whether they could act as carbon or nitrogen sources to support CRE growth. Growth of CRE isolates on sole carbon or nitrogen sources were tested by inoculating each CRE isolate into a minimal medium supplemented with the nutrient of interest. First, CRE isolates were passaged from frozen glycerol stocks onto R2A plates. Cells were suspended in M9 minimal media to a turbidity of 42% using a Turbidimeter (Biolog Inc., Hayward, CA), then further diluted 1 in 6 to inoculate the carbon or nitrogen utilisation assays. M9 minimal media contained the following (at final concentrations): 22 mM $KH_2PO_4$, 42 mM $Na_2HPO_4$, 9 mM NaCl, 0.49 mM $MgSO_4·7H_2O$, 0.09 mM $CaCl_2$, 0.011 mM $FeSO_4·7H_2O$. For the carbon assays the M9 minimal medium was also supplemented with 0.1% (w/v) $NH_4Cl$ (as the nitrogen source) and the carbon sources were individually tested at 0.5% (w/v) concentration. For the nitrogen assays the M9 minimal medium was also supplemented with 0.5% (w/v) glucose (as the carbon source) and the nitrogen sources were individually tested at 0.1% (w/v) concentration. Cultures were incubated anaerobically and aerobically in a plate reader and optical density ($OD_{600}$) readings were collected every 15 min for 24 h.

### Mixed nutrient preference assays
CRE isolates were passaged from frozen glycerol stocks on R2A plates. Cells were suspended in M9 minimal media to a turbidity of 42% using a Turbidimeter, and then further diluted 1 in 6 to inoculate the nutrient preference assays. M9 minimal media was supplemented with 0.015% (w/v) of each of the following nutrients (as the sole carbon and nitrogen sources): L-arabinose, D-fructose, L-fucose, D-galactose, D-glucose, D-mannose, D-ribose, D-xylose, N-acetylglucosamine, D-maltose, sucrose, D-trehalose, L-alanine, L-arginine, L-aspartate, L-glutamate, L-glycine, L-isoleucine, L-leucine, L-lysine, L-methionine, L-phenylalanine, L-proline, L-threonine, L-tryptophan, L-tyrosine, L-valine, uracil and succinate. Cultures were incubated anaerobically and aerobically in a plate reader and $OD_{600}$ readings were collected every 15 min for 24 h. Cultures were also incubated anaerobically and aerobically for 24 h and samples were collected at 0, 4, 8, and 24 h post-inoculation for ¹H-NMR spectroscopy. The starting nutrient concentration was

measured by the integration of a representative peak in the NMR spectrum from samples collected at 0 h and this concentration was set to 100% to enable measurements of percent nutrient remaining at 4, 8 and 24 h.

## Analysis of Microbial Growth Assays (AMiGA)

Growth curves were analysed in Python (v3.6.5) using the AMiGA software (available at https://github.com/firasmidani/amiga)[86]. Growth of each isolate was measured with six replicates from 2 to 3 independent experiments. Growth curves were normalised to the no substrate control using the subtraction method. AMiGA performed Gaussian Process regression to test differential growth between the substrate and the no substrate control. The functional difference in the $OD_{600}$ (and its credible interval) was also computed between the substrate and the no substrate control.

## Metabolite measurements from healthy human faeces

The concentration of 10 microbial metabolites were measured from the faeces of 12 healthy human donors (5 males, 7 females) by $^1$H-NMR spectroscopy. This was a separate set of faecal samples than the faecal samples that were used to inoculate the faecal culture experiments. Female faecal donors were $30.9 \pm 12.1$ years old and male donors were $27.8 \pm 5.3$ years old. Metabolite concentrations were measured from NMR spectra using the Chenomx NMR Suite (v8.6). Table S2 shows the approximate minimum, average, and maximum concentrations of each metabolite that was tested in the metabolite inhibition assays, which were based on the measurements of the microbial metabolites from the healthy donor faeces.

## Mixed metabolite inhibition assay

Because healthy faecal microbiota have intact colonisation resistance that prevents or restricts CRE growth, compounds that were present in antibiotic-naïve faecal microbiota but decreased in antibiotic-treated faecal microbiota were tested as potential inhibitory compounds in growth assays. We tested whether a mixture of microbial metabolites (that were decreased in antibiotic-treated faecal microbiota) would be able to inhibit the growth of each CRE isolate. The metabolite mixture was composed of formate, acetate, propionate, butyrate, valerate, isobutyrate, isovalerate, lactate, 5-aminovalerate and ethanol. Each metabolite was added to the mixture at a concentration that mimicked the average concentration measured in healthy human faecal samples (Table S2). Luria broth (LB) was supplemented with the metabolite mixture and compared to a no metabolite control. Overnight cultures of each CRE isolate were inoculated into the supplemented LB at $10^3$ CFU/ml. The assay was run at pH 6, 6.5 and 7 to mimic the pH range found in healthy faeces[24,87]. Cultures were incubated anaerobically for 16 h at 37 °C. Growth was measured at $OD_{600}$ after 16 h of incubation. This experiment was also repeated by supplementing the metabolite mixture into M9 minimal medium that was supplemented with 0.015% (w/v) of each of the following nutrients (as the sole carbon and nitrogen sources): L-arabinose, D-fructose, L-fucose, D-galactose, D-glucose, D-mannose, D-ribose, D-xylose, N-acetylglucosamine, D-maltose, sucrose, D-trehalose, L-alanine, L-arginine, L-aspartate, L-glutamate, L-glycine, L-isoleucine, L-leucine, L-lysine, L-methionine, L-phenylalanine, L-proline, L-threonine, L-tryptophan, L-tyrosine, L-valine, uracil and succinate. Overnight cultures of each CRE isolate were inoculated into the supplemented M9 medium at $10^3$ CFU/ml and the assay was run at pH 6, 6.5 and 7. Cultures were incubated anaerobically for 16 h at 37 °C. Growth was measured at $OD_{600}$ after 16 h of incubation.

We also measured the growth of CRE isolates in LB supplemented with a mixture of acetate, propionate, butyrate and valerate. Each metabolite was added to the mixture at a concentration that mimicked the average concentration measured in healthy human faecal samples (Table S2). LB was supplemented with the metabolite mixture and compared to a no metabolite control. The assay was run at pH 6.5.

Overnight cultures of each CRE isolate were inoculated into the supplemented LB at $10^3$ CFU/ml. Cultures were incubated anaerobically for 16 h at 37 °C. Growth was measured at $OD_{600}$ after 16 h of incubation.

## Individual metabolite inhibition assay

We next tested whether individual microbial metabolites (that were decreased in antibiotic-treated faecal microbiota) would be able to inhibit the growth of each CRE isolate. The individual metabolites tested included formate, acetate, propionate, butyrate, valerate, isobutyrate, isovalerate, lactate, 5-aminovalerate, and ethanol. Each metabolite was tested at three concentrations that mimicked the low, average, and high concentrations measured in healthy human faecal samples (Table S2). LB was supplemented with each metabolite and compared to a no metabolite control. Overnight cultures of each CRE isolate were inoculated into the supplemented LB at $10^3$ CFU/ml. The assay was run at pH 6.5 as this was representative of the distal gut and showed inhibition in the mixed metabolite assay. Cultures were incubated anaerobically for 16 h at 37 °C. Growth was measured at $OD_{600}$ after 16 h of incubation.

## Mouse model of intestinal colonisation with carbapenem-resistant *E. coli*

Ethical approval for mouse experiments was received from the Imperial College London Animal Welfare and Ethical Review Body (PF93C158E) and performed under the authority of the UK Home Office, as described in the Animals (Scientific Procedures) Act 1986. We abided by standards outlined in the Animal Research: Reporting of In Vivo Experiments (ARRIVE) guidelines.

Eight- to ten-week-old female wild-type C57BL/6 mice (purchased from Envigo, Huntingdon, UK) were acclimatised for 1 week prior to the start of the experiment. Each test group contained five mice that were housed five per cage (in individually ventilated cages). Mice were provided autoclaved food (RM1, Special Diet Services, Essex, UK), water (provided ad libitum), and bedding (Aspen chip 2 bedding, NEPCO, Warrensburg, New York). Mice were maintained at 20–22 °C and 45–65% humidity in a 12-h light and 12-h dark cycle (all interventions were performed during the light cycle).

We performed a mouse experiment to measure changes in nutrient availability in mouse faeces following TZP treatment and carbapenem-resistant *E. coli* colonisation. We also measured carbapenem-resistant *E. coli* growth in mouse faeces following the administration of a mixture of inhibitory metabolites. We used a previously published mouse model of intestinal CRE colonisation as described by Perez and colleagues (Figs. 7a, 9a)[19]. The dosing schedule of the metabolite mixture was designed based on a previous study from our group, where we investigated the effects of the microbial metabolite valerate on the intestinal colonisation with *C. difficile* in a mouse model[7].

In this study, mice were subcutaneously administered 8 mg of TZP (in 0.2 ml saline) or 0.2 ml of saline once per day for 5 days to mimic the daily TZP dose recommended for human adults[19]. On the third day of TZP or saline administration, $10^6$ CFU of *E. coli* ST617 (diluted in 0.2 ml saline) were orally administered to the mice. Next, TZP-treated mice that were colonised with *E. coli* were fed by oral gavage with 0.2 ml of a mixture of metabolites (glycerol triacetate, 65 mM; glycerol tripropionate, 15 mM; glycerol tributyrate, 15 mM; and glycerol trivalerate, 3.5 mM; diluted in PBS) or 0.2 ml of PBS once per day for 3 days. Faecal samples were collected for carbapenem-resistant *E. coli* plate counts (on Brilliance CRE agar, Oxoid, Basingstoke, UK) and for $^1$H-NMR spectroscopy.

## Statistical analysis

Statistical tests were performed using IBM SPSS Statistics Software 27 (IBM Corp, Armonk, New York), GraphPad Prism 9.4.1 (La Jolla, California), or in R (v4.2.2). A p value less than 0.05 was considered

significant. 16S rRNA gene sequencing data (weighted as absolute abundances) were analysed using Wilcoxon signed rank test (two-sided) with Benjamini & Hochberg FDR correction using the DA.wil function within in the DAtest R package version 2.7.18. Peak integration values from the $^1$H-NMR data were analysed using Wilcoxon signed rank test (two-sided) with Benjamini & Hochberg FDR correction using the DA.wil function within in the DAtest R package version 2.7.18 or a paired t-test (two-sided) with Benjamini & Hochberg FDR correction using the pairwise_t_test function within the rstatix R package version 0.7.1. Multivariate general linear models were analysed in IBM SPSS Statistics Software 27 to determine whether faecal donor age or sex impacted the $^1$H-NMR spectroscopy or 16S rRNA gene sequencing measurements, using the $^1$H-NMR metabolite measurements or 16S rRNA gene sequencing read counts as the dependent variables, sex as a fixed factor, and age as a covariate. Regularised canonical correlation analysis (rCCA) was used to correlate the 16S rRNA gene sequencing data and $^1$H-NMR data using the mixOmics R package version 6.22.0[7,21]. Partial Spearman correlations were calculated between the $^1$H-NMR spectroscopy data and 16S rRNA gene sequencing data from the faecal culture experiments using IBM SPSS Statistics Software 27 and plotted using the corrplot function within the corrplot R package version 0.92. A paired t-test (two-sided) was used to compare log10-transformed CRE counts from TZP-treated human faecal cultures compared to water-treated human faecal cultures. For the carbon and nitrogen utilisation assays the growth curves were analysed in Python (v3.6.5) using the AMiGA software (available at https://github.com/firasmidani/amiga)[86]. For the nutrient preference experiment changes in the concentration of nutrients and metabolites for each CRE isolate were compared to the sterile media control over time using a two-way mixed ANOVA followed by pairwise comparisons with Bonferroni correction. An unpaired t-test (two-sided) was used to compare log10-transformed E. coli counts from the faecal pellets of TZP-treated mice compared to saline-treated mice. For the in vitro mixed metabolite inhibition test, growth of the CRE isolate in the metabolite mixture was compared to growth in the no metabolite control at each pH using an unpaired t-test (two-sided). For the in vitro individual metabolite inhibition test, growth of the CRE isolate in the presence of each metabolite was compared to the no metabolite control using a one-way ANOVA followed by Dunn's multiple comparison test. A Mann–Whitney U test (two-sided) was used to compare log10-transformed E. coli counts from the faecal pellets of metabolite-treated mice compared to PBS-treated mice.

### Reporting summary

Further information on research design is available in the Nature Portfolio Reporting Summary linked to this article.

## Data availability

Source data are provided with this paper. 16S rRNA gene sequencing, 16S rRNA gene qPCR, and $^1$H-NMR datasets generated in this study have been deposited into the Figshare repository at https://doi.org/10.6084/m9.figshare.21948233. Raw reads for whole genome sequences for the CRE patient isolates have been deposited to the European Nucleotide Archive (ENA) under the BioProject PRJEB60914 with isolate accession numbers NDM-5 E. coli ST617 (ERR11452083), NDM-1 K. pneumoniae ST1026 (ERR11452082), and NDM-1 E. hormaechei ST278 (ERR11452081) and strains are available upon request. Raw reads for the 16S rRNA gene sequencing data have also been deposited to the ENA under the BioProject PRJEB60914 with sample accession numbers and links found in the Fig. 1 Source Data file. For the whole genome sequencing data from the CRE patient isolates, the Kraken bacterial database is available at http://ccb.jhu.edu/software/kraken2/, the ResFinder database is available at https://cge.food.dtu.dk/services/ResFinder/, and the MLST databases are available at https://pubmlst.

org and https://bigsdb.pasteur.fr/klebsiella/. For the 16S rRNA gene sequencing data the SILVA bacterial database version 138.1 is available at https://www.arb-silva.de/. For the $^1$H-NMR spectroscopy data the compound database in the Chenomx NMR Suite v9.02 is available at https://www.chenomx.com/. Source data are provided with this paper.

## Code availability

NMR spectra were imported into MATLAB r2019b and processed using a custom script available in the Figshare repository at https://doi.org/10.6084/m9.figshare.21948233 using the code available at the Zenodo repository at https://doi.org/10.5281/zenodo.3077413.

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

## Acknowledgements

This work was supported by funding awarded to J.A.K.M.: a Medical Research Council (MRC) New Investigator Research Grant (MR/W025655/1), a Wellcome Trust Institutional Strategic Support Fund Springboard Fellowship, and start-up funds from the Department of Life Sciences at Imperial College London. E.J. is a Rosetrees/Stoneygate 2017 Imperial College Research Fellow, funded by Rosetrees Trust and the Stoneygate Trust (Fellowship no. M683). E.J. is affiliated with the National Institute for Health Research Health Protection Research Unit (NIHR HPRU) in Healthcare Associated Infections and Antimicrobial Resistance at Imperial College London in partnership with the UK Health Security Agency (previously PHE), in collaboration with, Imperial Healthcare Partners, University of Cambridge and University of Warwick. F.J.D. received funding from the Medical Research Council (MRC) Clinical Academic Research Partnership Scheme (MR/T005254/1). T.B.C. received funding from a UKRI Impact Accelerator Award (MR/X502959/1). B.H.M. is the recipient of an NIHR Academic Clinical Lectureship (CL-2019-21-002). The Division of Digestive Diseases received financial and infrastructure support from the National Institute for Health Research (NIHR) Imperial Biomedical Research Centre (BRC) based at Imperial College Healthcare NHS Trust and Imperial College London. This publication made use of the PubMLST website (https://pubmlst.org/) developed by Keith Jolley and sited at the University of Oxford[88]. The development of the PubMLST website was funded by the Wellcome Trust.

## Author contributions

A.Y.G.Y.—Formal analysis: equal; investigation: lead; methodology: equal; visualisation: equal; writing—review & editing: supporting. O.G.K.—Investigation: supporting; writing—review & editing: supporting. O.O.—Formal analysis: supporting; investigation: supporting. S.K.—Formal analysis: supporting; investigation: supporting. V.H.—Formal analysis: supporting; investigation: supporting; writing – review &

editing: supporting. P.M.—Formal analysis: supporting; investigation: supporting. N.D.—Investigation: supporting. R.G.—Resources: supporting, writing—review & editing: supporting. G.S.—Resources: supporting, writing—review & editing: supporting. E.J.—Formal analysis: supporting; investigation: supporting; resources: supporting; writing—review & editing: supporting. F.J.D.—Resources: supporting; writing—review & editing: supporting. T.B.C.—Investigation: supporting; methodology: supporting; writing – review & editing: supporting. B.H.M.—Resources: supporting; writing—review & editing: supporting. J.R.M.—Resources: supporting; writing – review & editing: supporting. J.A.K.McD.—Conceptualisation: lead; data curation: lead; formal analysis: equal; funding acquisition: lead; methodology: equal; project administration: lead; resources: lead; supervision: lead; visualisation: equal; writing—original draft: lead; writing—review & editing: lead.

## Competing interests

J.A.K.M., A.Y.G.Y. and O.G.K. have filed a patent application related to this work (patent application number 2217266.2). B.H.M. received consultancy fees from Finch Therapeutics Group, Ferring Pharmaceuticals, and Summit Therapeutics, outside of the submitted work. J.R.M. received consultancy fees from EnteroBiotix Ltd. and Cultech Ltd, outside of the submitted work. The remaining authors declare no competing interests.

## Additional information

[1]Centre for Bacterial Resistance Biology, Department of Life Sciences, Imperial College London, London SW7 2AZ, UK. [2]Centre for Bacterial Resistance Biology, Department of Infectious Disease, Imperial College London, London SW7 2AZ, UK. [3]Division of Digestive Diseases, Department of Metabolism, Digestion and Reproduction, Faculty of Medicine, St Mary's Hospital Campus, Imperial College London, London, UK. [4]Department of Infectious Disease, Imperial College Healthcare NHS Trust, London, UK. [5]UCL Centre for Clinical Microbiology, University College London, London, UK. [6]NIHR Health Protection Research Unit in Healthcare Associated Infections and Antimicrobial Resistance, Department of Infectious Disease, Imperial College London, London, UK. [7]Department of Infectious Disease Epidemiology, School of Public Health, Imperial College London, London, UK. [8]Departments of Gastroenterology and Hepatology, St Mary's Hospital, Imperial College Healthcare NHS Trust, Paddington, London, UK. ✉e-mail: julie.mcdonald@imperial.ac.uk

