## [Peer Review File · Nature Communications]

Antibiotics promote intestinal growth of carbapenem-resistant Enterobacteriaceae by enriching nutrients and depleting microbial metabolitesEditorial Note: This manuscript has been previously reviewed at another journal that is not operating a transparent peer review scheme. This document only contains reviewer comments and rebuttal letters for versions considered at *Nature Communications*.

Reviewer #2 (Remarks to the Author):

In this revised manuscript, the authors have thoughtfully and quite comprehensively addressed the major issues raised by the reviewers. The manuscript is much improved and the message that antibiotic treatment increases nutrient concentrations that support the growth of CRE bacterial species while also decreasing inhibitory metabolites is interesting and of potential clinical relevance. Most of the responses to reviewers issues were sensible and reasonable, but Comment 4 to Reviewer 2 which states: "However, it is important to replicate studies to ensure the results are reproducible" does not convince this reviewer that not repeating an experiment is well justified. Finally, in the discussion the authors imply that little is known about mechanisms by which metabolites such as SCFAs inhibit Enterobacteriaceae. There was an interesting recent paper (PMID: 35120663) implicating Acyl-CoA thioesterase activity while Sorbara et al. demonstrated intracellular acidification by SCFAs that the authors may want to mention in the discussion.

Reviewer #3 (Remarks to the Author):

I would like to congratulate the authors on the revised manuscript. I has been carefully adapted and now presents a more balanced manuscript without unnecessary overinterpretation.

Reviewer #4 (Remarks to the Author):

This is a revised manuscript submitted to Nature Communications. Although significant improvement has been made by the authors, further clarification on the data analysis is needed.

1. Author response: effects of age and sex on faecal microbiome for the faecal metabolite measurements

The authors used a multivariate general linear model to analyze the data to determine whether age or sex of the donors could have impacted these measurements and stated that no significant differences in any of the measured metabolites due to age or sex according to this model. The reviewer didn't see any data on this in the revision. The authors should provide the methods used and the results obtained in the revision, at least in the supplementary materials.

2. "Author response: effects of age and sex on faecal microbiome for the faecal culture experiments (data represented in Figs. 1a, 1b, 2, & S1)

As discussed above, in the faecal culture experiments faeces from each donor was used to seed all test conditions (including the control condition). Because the same faecal microbiota was tested under all experimental conditions (for each of the 8 antibiotics and for the antibiotic-naïve control) this means that there is no difference in the age or sex of the donor faeces used to seed faecal cultures in each experimental group. Therefore, for the faecal culture experiments age or sex of the donor would not impact the outcomes of this experiment, and we do not believe that it is necessary to conduct an analysis to determine whether there was a difference in the faecal microbiota between different donors in each group due to sex or age. "

In view of the small sample size, it is important and necessary to consider sex and age in examining the effects of faecal microbiota, for difference and correlation analysis. Please use some sort of multivariate models (e.g. GLM) in all possible cases.

3. NMR spectra underwent auto processing, including exponential window function (line broadened by 0.3 Hz), auto phasing (zero-order), auto baseline correction, and auto referencing. Here please provide the details of the software used and parameters set during the analysis.

4. "Reviewer 1 Comment 6:

[Statistics continued:] As I mentioned below in the validity section, the sample size for this study is quite small. The authors should comment on this with respect to the validity of the results. More importantly, they should include some sort of limitation statement in the discussion that clearly states that their sample size was small and these results need to be assessed on a much larger

cohort. The authors also need to ensure that the age and sex of the healthy participants did not affect the structure (16S rRNA) or functioning (metabolite concentrations) of the healthy faecal samples.

Reviewer 3 Comment 6:

Line 207ff: The association between changes in metabolites and composition is a very interesting aspect of the analysis, but the exploration is not presented very well. It is unclear which associations are statistically significant and robust over multiple donors. This aspect of the analysis should be expanded and could then be included also in a main figure."

In the revision, the authors further clarified that "Regularised canonical correlation analysis (rCCA) is an unsupervised exploratory approach used to analyse the correlation structure between two multivariate datasets that have been acquired from the same samples^{22,23}."

Here, CCA cannot provide significance and cannot account covariates such as age and sex. The reviewer's suggestion is to use both multivariate (e.g. CCA) and univariate (e.g. Partial spearman) methods for the correlation analysis.

RESPONSE TO REVIEWER COMMENTS

We would like to thank the reviewers for giving their time and effort to review our manuscript and our previous response to the reviewer comments. We have outlined below the further improvements we made to our manuscript based on the additional reviewer comments.

REVIEWER #2 (REMARKS TO THE AUTHOR):

Reviewer 2 Comment 1:

In this revised manuscript, the authors have thoughtfully and quite comprehensively addressed the major issues raised by the reviewers. The manuscript is much improved and the message that antibiotic treatment increases nutrient concentrations that support the growth of CRE bacterial species while also decreasing inhibitory metabolites is interesting and of potential clinical relevance.

Author response:

Thank you.

Reviewer 2 Comment 2:

Most of the responses to reviewers issues were sensible and reasonable, but Comment 4 to Reviewer 2 which states: "However, it is important to replicate studies to ensure the results are reproducible" does not convince this reviewer that not repeating an experiment is well justified.

Author response:

The mouse work performed in this manuscript was designed using power calculations to achieve the desired statistical power, and our *in vivo* results support observations from our *ex vivo* and *in vitro* data sets. Moreover, we would like to highlight the considerable ethical concerns and restrictions related to the unnecessary use of animals in research and the ethical requirement that researchers must reduce animal use in studies where possible.

However, to address the reviewer's comment we have revised the supplementary discussion section to further discuss the limitation of the lack of repetition of the mouse experiments. In particular, we have (1) more clearly stated that this is a limitation of the mouse experiments, (2) highlighted the 3Rs principle of reduction of animal use, (3) highlighted that the *in vivo* mouse work is not a stand-alone data set and that we have *ex vivo* and *in vitro* data that supports our *in vivo* data (see lines 127-139):

"Finally, a limitation of the mouse experiments is that they were performed as one independent experiment. However, our mouse experiments were designed using power calculations to achieve the desired statistical power. Designing an experiment to achieve a desired power (e.g. $\alpha = 0.05$ and a power of 0.80) and repeating this same experimental design several times may result in a higher power than is needed and overpower the study, resulting in the use of more animals than is needed¹⁶. In accordance with the 3Rs principle of reduction, we minimised animal use to obtain sufficient data to answer our research question¹⁷. Our *in vivo* results are not a stand-alone data set, as they are substantiated by complementary data from our *ex vivo* and *in vitro* experiments that provide strong support for our *in vivo* data. Future studies that continue the development of this metabolite mixture as a new treatment could repeat this experiment while also generating new data to make the most of using these animals for experimentation (e.g. testing multiple doses of the metabolite mixture)^{18,19}."

Reviewer 2 Comment 3:

Finally, in the discussion the authors imply that little is known about mechanisms by which metabolites such as SCFAs inhibit Enterobacteriaceae. There was an interesting recent paper (PMID: 35120663) implicating Acyl-CoA thioesterase activity while Sorbara et al. demonstrated intracellular acidification by SCFAs that the authors may want to mention in the discussion.

Author response:

We have added in discussions of these two papers into the discussion section (see lines 688-708):

“There have been some investigations into the mechanisms that acetate, propionate, and butyrate use to inhibit bacterial growth. A previous study by Sorbara and colleagues demonstrated that acetate, propionate, and butyrate inhibited *E. coli* and *K. pneumoniae* growth through intracellular acidification¹⁷. They showed that these three SCFAs (present in their nonionised form at low pH) diffused across the bacterial membrane into the cytoplasm. Once inside the bacterial cell these SCFAs dissociated into their ionised forms, releasing protons into the cytoplasm and acidifying the intracellular pH. In another study Park and colleagues demonstrated that butyrate exhibited strain-dependent inhibitory activity against Bacteroidales, which was impacted by the utilisation of distinct sugars in a context-dependent manner⁵⁴. They also demonstrated that variation in Acyl-CoA thioesterase and transferase activity governed differences to butyrate resistance in *Bacteroides*. However, in this study we demonstrated that additional metabolites (valerate, isobutyrate, isovalerate, ethanol) also caused growth inhibition of CRE, and further research is required to solve the mechanisms that these inhibitory metabolites use to restrict intestinal CRE colonisation. Moreover, it is unclear whether these 7 inhibitory metabolites may act using one mechanism of inhibition or whether they may mediate inhibition by multiple different mechanisms. We saw differences in the metabolites that inhibited the growth of the *E. coli*, *K. pneumoniae*, and *E. hormaechei* strains tested, and it is not clear whether there are differences in the mechanisms in these three different species. It is also unclear whether these metabolites may have synergistic inhibitory effects, or whether these mechanisms may be impacted by different gut environmental conditions. Future studies are needed to investigate how these metabolites impact pathogens, other members of the gut microbiota, and the host to provide colonisation resistance against CRE.”

REVIEWER #3 (REMARKS TO THE AUTHOR):

Reviewer 3 Comment 1:

I would like to congratulate the authors on the revised manuscript. It has been carefully adapted and now presents a more balanced manuscript without unnecessary overinterpretation.

Author response:

Thank you.

REVIEWER #4 (REMARKS TO THE AUTHOR):

This is a revised manuscript submitted to Nature Communications. Although significant improvement has been made by the authors, further clarification on the data analysis is needed.

Reviewer 4 Comment 1:

1. Author response: effects of age and sex on faecal microbiome for the faecal metabolite measurements

The authors used a multivariate general linear model to analyze the data to determine whether age or sex of the donors could have impacted these measurements and stated that no significant differences in any of the measured metabolites due to age or sex according to this model. The reviewer didn't see any data on this in the revision. The authors should provide the methods used and the results obtained in the revision, at least in the supplementary materials.

Author response:

We have added this additional analysis to the methods and supplementary results as recommended by the reviewer:

For methods see lines 1036-1040:

“Multivariate general linear models were analysed in IBM SPSS Statistics Software 27 to determine whether faecal donor age or sex impacted the ¹H-NMR spectroscopy or 16S rRNA gene sequencing measurements, using the ¹H-NMR metabolite measurements or 16S rRNA gene sequencing read counts as the dependent variables, sex as a fixed factor, and age as a covariate.”

For supplementary results see lines 60-64:

“A multivariate general linear model was also used to analyse the ¹H-NMR spectroscopy data from the donor faecal samples (data presented in **Table S1**) to determine whether age or sex of the donors impacted these measurements. There were no significant differences in any of the measured metabolites due to age or sex according to the multivariate general linear model (**Table S7**).”

Reviewer 4 Comment 2:

2. “Author response: effects of age and sex on faecal microbiome for the faecal culture experiments (data represented in Figs. 1a, 1b, 2, & S1)

As discussed above, in the faecal culture experiments faeces from each donor was used to seed all test conditions (including the control condition). Because the same faecal microbiota was tested under all experimental conditions (for each of the 8 antibiotics and for the antibiotic-naïve control) this means that there is no difference in the age or sex of the donor faeces used to seed faecal cultures in each experimental group. Therefore, for the faecal culture experiments age or sex of the donor would not impact the outcomes of this experiment, and we do not believe that it is necessary to conduct an analysis to determine whether there was a difference in the faecal microbiota between different donors in each group due to sex or age.”

In view of the small sample size, it is important and necessary to consider sex and age in examining the effects of faecal microbiota, for difference and correlation analysis. Please use some sort of multivariate models (e.g. GLM) in all possible cases.

Author response:

To further clarify our previous response to the reviewer comments, the data presented for the faecal culture experiments consists of matched donor samples between the different treatment groups. In other words, for each donor a single faecal sample was homogenised and split to seed the antibiotic-naïve group and each antibiotic-treated group (so one faecal sample was used to inoculate 9 different test conditions). Designing the experiment in this manner means that we have identical faecal samples used to seed each experimental group, and therefore the donor characteristics (age and sex) are identical in each group (similar to experiments that have collected samples from participants pre-intervention and post-intervention). Therefore, in this matched experimental design we have already controlled for the age and sex of the donors.

However, we have added multivariate general linear models for the ¹H-NMR spectroscopy data and 16S rRNA gene sequencing data acquired from the antibiotic-naïve faecal culture samples. We have added this information to the supplementary results (see lines 48-58):

“Multivariate general linear models were used to analyse the ¹H-NMR spectroscopy data and 16S rRNA gene sequencing data from the antibiotic-naïve faecal cultures to determine whether age or sex of the donors impacted these measurements. We found no significant differences in any of the measured metabolites due to age or sex (**Table S5**). We found no significant differences in any of the bacterial families due sex, however we found one bacterial family (*Muribaculaceae*) was affected by age according to this model (**Table S6**). However, *Muribaculaceae* was only detected in 1 of the 11 donors, and it was present in this donor at a very low level approaching the limit of detection. Moreover, we did not find a significant difference in *Muribaculaceae* abundance in any of the antibiotic-treated groups (**Fig. 1A**). Therefore, the potential influence of

age on *Muribaculaceae* abundance did not impact the outcomes of this study.”

Reviewer 4 Comment 3:

3. NMR spectra underwent auto processing, including exponential window function (line broadened by 0.3 Hz), auto phasing (zero-order), auto baseline correction, and auto referencing. Here please provide the details of the software used and parameters set during the analysis.

Author response:

We have revised the methods section of the manuscript to include the requested details (see lines 872-874):

“All NMR spectra were processed in Topspin (v3.2.6) in the same manner, including an exponential window function (line broadened by 0.3Hz), zero-order auto phasing (using command apk0), baseline correction (using command abs), and referencing with TSP (using command sref).”

Reviewer 4 Comment 4:

4. “Reviewer 1 Comment 6:

[Statistics continued:] As I mentioned below in the validity section, the sample size for this study is quite small. The authors should comment on this with respect to the validity of the results. More importantly, they should include some sort of limitation statement in the discussion that clearly states that their sample size was small and these results need to be assessed on a much larger cohort. The authors also need to ensure that the age and sex of the healthy participants did not affect the structure (16S rRNA) or functioning (metabolite concentrations) of the healthy faecal samples.

Reviewer 3 Comment 6:

Line 207ff: The association between changes in metabolites and composition is a very interesting aspect of the analysis, but the exploration is not presented very well. It is unclear which associations are statistically significant and robust over multiple donors. This aspect of the analysis should be expanded and could then be included also in a main figure.”

In the revision, the authors further clarified that “Regularised canonical correlation analysis (rCCA) is an unsupervised exploratory approach used to analyse the correlation structure between two multivariate datasets that have been acquired from the same samples^{22,23}.”

Here, CCA cannot provide significance and cannot account covariates such as age and sex. The reviewer’s suggestion is to use both multivariate (e.g. CCA) and univariate (e.g. Partial spearman) methods for the correlation analysis.

Author response:

As suggested by reviewer 4 above, we have added univariate correlations (partial Spearman) analysis to complement the multivariate rCCA correlation plots.

We have revised the methods section to include this new analysis (see lines 1042-1045):

“Partial Spearman correlations were calculated between the ¹H-NMR spectroscopy data and 16S rRNA gene sequencing data from the faecal culture experiments using IBM SPSS Statistics Software 27 and plotted using the corplot function within the corplot R package version 0.92.”

We added this new analysis into our results section (see lines 285-288):

“Partial Spearman correlations were also calculated to determine significant correlations between the ¹H-NMR spectroscopy data and 16S rRNA gene sequencing data and showed similar results to the data presented in the correlation circle plots (**Figs. S2-S9**).”